# Fast site-to-site electron transfer of high-entropy alloy nanocatalyst driving redox electrocatalysis

Hongdong Li[1], Yi Han[1], Huan Zhao[1], Wenjing Qi[2], Dan Zhang[1,3], Yaodong Yu[1], Wenwen Cai[1], Shaoxiang Li[3], Jianping Lai[1✉], Bolong Huang [4✉] & Lei Wang [1,3✉]

Designing electrocatalysts with high-performance for both reduction and oxidation reactions faces severe challenges. Here, the uniform and ultrasmall (~3.4 nm) high-entropy alloys (HEAs) $Pt_{18}Ni_{26}Fe_{15}Co_{14}Cu_{27}$ nanoparticles are synthesized by a simple low-temperature oil phase strategy at atmospheric pressure. The $Pt_{18}Ni_{26}Fe_{15}Co_{14}Cu_{27}$/C catalyst exhibits excellent electrocatalytic performance for hydrogen evolution reaction (HER) and methanol oxidation reaction (MOR). The catalyst shows ultrasmall overpotential of 11 mV at the current density of 10 mA cm$^{-2}$, excellent activity (10.96 A mg$^{-1}$_Pt at −0.07 V vs. reversible hydrogen electrode) and stability in the alkaline medium. Furthermore, it is also the efficient catalyst (15.04 A mg$^{-1}$_Pt) ever reported for MOR in alkaline solution. Periodic DFT calculations confirm the multi-active sites for both HER and MOR on the HEA surface as the key factor for both proton and intermediate transformation. Meanwhile, the construction of HEA surfaces supplies the fast site-to-site electron transfer for both reduction and oxidation processes.

[1] Key Laboratory of Eco-chemical Engineering, Key Laboratory of Optic-electric Sensing and Analytical Chemistry of Life Science, Taishan Scholar Advantage and Characteristic Discipline Team of Eco Chemical Process and Technology, College of Chemistry and Molecular Engineering, Qingdao University of Science and Technology, 266042 Qingdao, P. R. China. [2] College of Chemistry, Chongqing Normal University, 401331 Chongqing, P. R. China. [3] Shandong Engineering Research Center for Marine Environment Corrosion and Safety Protection, College of Environment and Safety Engineering, Qingdao University of Science and Technology, 266042 Qingdao, P. R. China. [4] Department of Applied Biology and Chemical Technology, The Hong Kong Polytechnic University, Hung Hom, Kowloon, Hong Kong SAR, China. ✉email: jplai@qust.edu.cn; bhuang@polyu.edu.hk; inorchemwl@126.com

Electrocatalytic processes play a vital role in energy conversion and reducing the environmental pollution[1–5]. To improve the activity, selectivity, and stability of the catalytic reaction, it is necessary to develop high-performance advanced catalysts that can meet the needs of rapid development[6–12]. High-entropy alloys (HEAs) have attracted wide interest as catalytic materials in the past few years[13–16]. The alloy contains five or more elements which have similar atomic ratios[16,17]. The atomic size of each component is different, which can cause lattice distortion[18]. Besides, the presence of multiple components is conducive to promoting the formation of the solid solution phase and inhibiting the movement of dislocations. These characteristics of HEA lead to some unique characteristics, such as strong fracture toughness, corrosion resistance, and high mechanical strength[19–21]. The ultimate goal of HEA is to adapt these characteristics to any desired response by using almost infinite possible combinations of elements and modifying their composition. In catalysis field, the adsorption of molecules and intermediates species on the surface of the catalyst affects the catalytic activity[13,22]. These adsorption energies can be adjusted by alloying as compared to pure elements to increase catalytic activity[23–25]. Recently, some HEAs had used as catalysts for electrocatalytic reactions, which display superior stability and catalytic selectivity and activity compared with traditional alloys[13,26–30]. However, the traditional method mainly produce bulk HEAs rather than nanostructures[26,31–34]. Moreover, the preparation of uniform nanostructured HEAs with small size (<10 nm) currently requires specific equipment (fast heating/cooling rate, ~$10^5$ K per second), high temperature (~2000 kelvin), and high temperature resistant and conductive substrate (carbon nanofiber), such as carbonthermal shock method[35].

In this work, taking into account the high abundance of Ni, Fe, Co, and Cu and they can easily form solid solutions with Pt. These transition metal elements are selected because their similar atomic radius and lower heat of formation make them likely to form stable HEAs[20,28]. In addition, the DFT calculation provides a means to better understand the catalytic process and direct catalytic design[13]. Based on this, we synthesize the small size (~3.4 nm) and uniform HEA $Pt_{18}Ni_{26}Fe_{15}Co_{14}Cu_{27}$ nanoparticles (NPs) by a simple low-temperature oil phase synthesis method at atmospheric pressure. The $Pt_{18}Ni_{26}Fe_{15}Co_{14}Cu_{27}$/C catalyst displays a low overpotential (11 mV at 10 mA cm$^{-2}$), the activity of 10.96 A mg$^{-1}_{Pt}$ at −0.07 V vs. reversible hydrogen electrode (RHE) for HER (reduction reaction) in 1 M KOH solution, and through the chronoamperometric method and 10,000th CV tests, it indicates that the notable stability of the HEA catalyst. We further also measure the electrocatalytic activity of $Pt_{18}Ni_{26}Fe_{15}Co_{14}Cu_{27}$/C for MOR (oxidation reaction) at 1 M KOH + 1 M CH$_3$OH electrolyte, it also exhibited high mass activity (15.04 A mg$^{-1}_{Pt}$). Theoretical calculations reveal that each element in HEA displays different contributions for the electrocatalysis process, which promotes the site-to-site electron transfer and the stabilization of the intermediates. HEA showed the multi-active sites for both HER and MOR for achieving superior performance and stability.

## Results

### Synthesis and characterizations of PtNiFeCoCu HEA NPs.
PtNiFeCoCu HEA NPs were prepared through a simple one-pot oil phase synthesis method at 220 °C for 2 h. The inductively coupled plasma atomic emission spectroscopy (ICP-AES) result shows that the atomic ratio of Pt, Ni, Fe, Co, and Cu was 18:26:15:14:27 (Supplementary Table 1), the prepared NPs-HEA was named $Pt_{18}Ni_{26}Fe_{15}Co_{14}Cu_{27}$. As shown in Fig. 1a, b, the transmission electron microscopy (TEM) images of $Pt_{18}Ni_{26}$-$Fe_{15}Co_{14}Cu_{27}$ NPs display the morphology is uniform, and the diameter of $Pt_{18}Ni_{26}Fe_{15}Co_{14}Cu_{27}$ NPs is about 3.4 ± 0.6 nm (the

inset in Fig. 1a). The high-resolution TEM (HRTEM) (Fig. 1c) image of $Pt_{18}Ni_{26}Fe_{15}Co_{14}Cu_{27}$ NPs exhibits that the lattice spacing is 0.218 nm, corresponding to the (111) facet. The powder X-ray diffraction (PXRD) pattern (Supplementary Fig. 1a) suggests the fcc structure of the NPs-HEAs, the two peaks at around 41.1º and 47.8º, which can be assigned to the (111) and (200) facets. The position of the broad diffraction peak is significant shift compared with the diffraction peaks of pure Pt, Ni, Fe, Co, and Cu (Supplementary Fig. 1b), it shows that these elements have been introduced into the nanostructure to form HEAs structure. Figure 1d shows the elemental maps of NPs-HEAs, Pt, Ni, Fe, Co, and Cu elements are uniformly distributed in the HEA nanostructure.

The X-ray photoelectron spectroscopy (XPS) of $Pt_{18}Ni_{26}Fe_{15}$-$Co_{14}Cu_{27}$ NPs was tested, it also shows the presence of these Pt, Ni, Fe, Co, and Cu elements (Supplementary Fig. 2a), the atomic ratio of Pt, Ni, Fe, Co, and Cu was 21.2:27.4:15.2:13.7:22.5, further proving that the HEA was successfully prepared. As shown in Supplementary Fig. 2b, the Pt $4f_{7/2}$ and Pt $4f_{5/2}$ peaks are located at 71.3 and 74.7 eV, respectively. Pt mainly exists in the form of zero valences and a small amount of 2+ valence[29]. The Ni 2p spectra display the coexistence of Ni$^0$ (852.8 eV) and Ni$^{2+}$ (855.6 eV) and a satellite peak locate at 861.6 eV (Supplementary Fig. 2c), the content of Ni$^0$ is lower than the content of Ni$^{2+}$ because of the high chemical activity of Ni[28,29]. The Fe 2p spectra in Supplementary Fig. 2d shows two peaks at 711.6 eV and 724.8 eV, which can be attributed to Fe $2p_{3/2}$ and Fe $2p_{1/2}$, respectively. The two Co 2p peaks can be assigned to Co $2p_{3/2}$ (780.8 eV) and Co $2p_{1/2}$ (797.0 eV) (Supplementary Fig. 2e). The Cu 2p spectra (Supplementary Fig. 2f) also exhibits Cu$^0$ (931.7 eV) and Cu$^{2+}$ (933.5 eV) peaks coexist in the sample.

### Electrocatalytic performance tests toward HER.
In order to study the electrocatalytic performance of PtNiFeCoCu/C, we further treated with acetic acid to remove residual surfactants. From the FTIR spectrum (Supplementary Fig. 3), the acetic acid-treated sample (the purified PtNiFeCoCu/C), no obvious characteristic peaks corresponding to oleylamine and CTAC were observed[36]. The HER (reduction reaction) activity and stability of the $Pt_{18}Ni_{26}Fe_{15}Co_{14}Cu_{27}$/C catalyst were investigated by a series of electrochemical tests and compared with commercial Pt/C. The Pt size (about 3.0 nm, Supplementary Fig. 4) on the Pt/C catalyst is similar to the size of HEA particles, so the electrocatalytic activity can be reasonably compared based on surface Pt sites. Figure 2a shows the CV curves of $Pt_{18}Ni_{26}Fe_{15}Co_{14}Cu_{27}$/C and commercial Pt/C catalysts in N$_2$-saturated 1 M KOH at a scan rate of 20 mV s$^{-1}$. After activation, the linear sweep voltammetry (LSV) curve of the $Pt_{18}Ni_{26}Fe_{15}Co_{14}Cu_{27}$/C catalyst displays a low overpotential of 11 mV at the current density of 10 mA cm$^{-2}$ (normalized to the electrode area), which is far superior to commercial Pt/C catalyst (84 mV) (Fig. 2b). As shown in Fig. 2b, d, the area activity (normalized to the geometric area) of NPs-HEAs $Pt_{18}Ni_{26}Fe_{15}Co_{14}Cu_{27}$/C catalyst reaches 83.78 mA cm$^{-2}$ at −0.07 V vs. RHE, far more than area activity of commercial Pt/C catalyst (8.42 mA cm$^{-2}$ at −0.07 V vs. RHE). From the Fig. 2c (LSV curves, normalized to the Pt mass), the mass activity for commercial Pt/C catalyst and $Pt_{18}Ni_{26}Fe_{15}Co_{14}Cu_{27}$/C catalyst are 0.83 A mg$^{-1}_{Pt}$ and 10.96 A mg$^{-1}_{Pt}$ at −0.07 V vs. RHE (Fig. 2d), respectively. And the $Pt_{18}Ni_{26}Fe_{15}Co_{14}Cu_{27}$/C catalyst also exhibits higher HER performance in alkaline medium among the reported Pt-based catalysts and HEAs catalysts (Supplementary Table 7). As shown in Fig. 2e, the Tafel slopes of $Pt_{18}Ni_{26}$-$Fe_{15}Co_{14}Cu_{27}$/C and Pt/C are 30 mV dec$^{-1}$ and 98 mV dec$^{-1}$, respectively, which demonstrate that the HEAs catalyst greatly boosts HER kinetics.

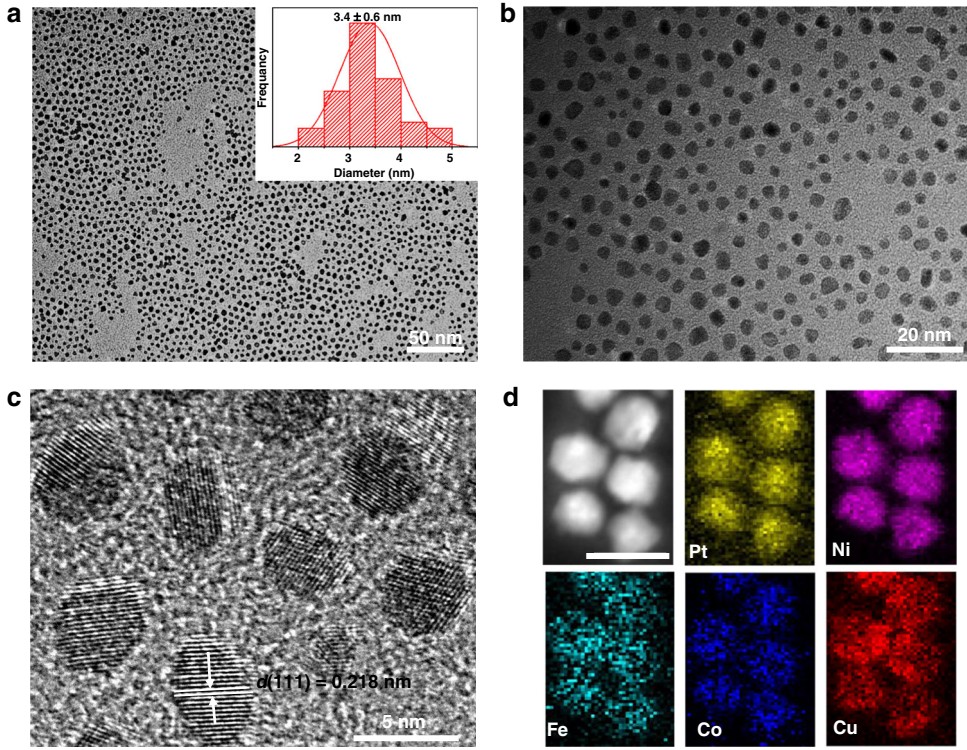

**Fig. 1 TEM images and elemental mapping analysis. a, b** TEM images of $Pt_{18}Ni_{26}Fe_{15}Co_{14}Cu_{27}$ nanoparticles. **c** HRTEM image of $Pt_{18}Ni_{26}Fe_{15}Co_{14}Cu_{27}$ nanoparticles. **d** The corresponding elemental mapping of $Pt_{18}Ni_{26}Fe_{15}Co_{14}Cu_{27}$ nanoparticles (scale bar, 5 nm).

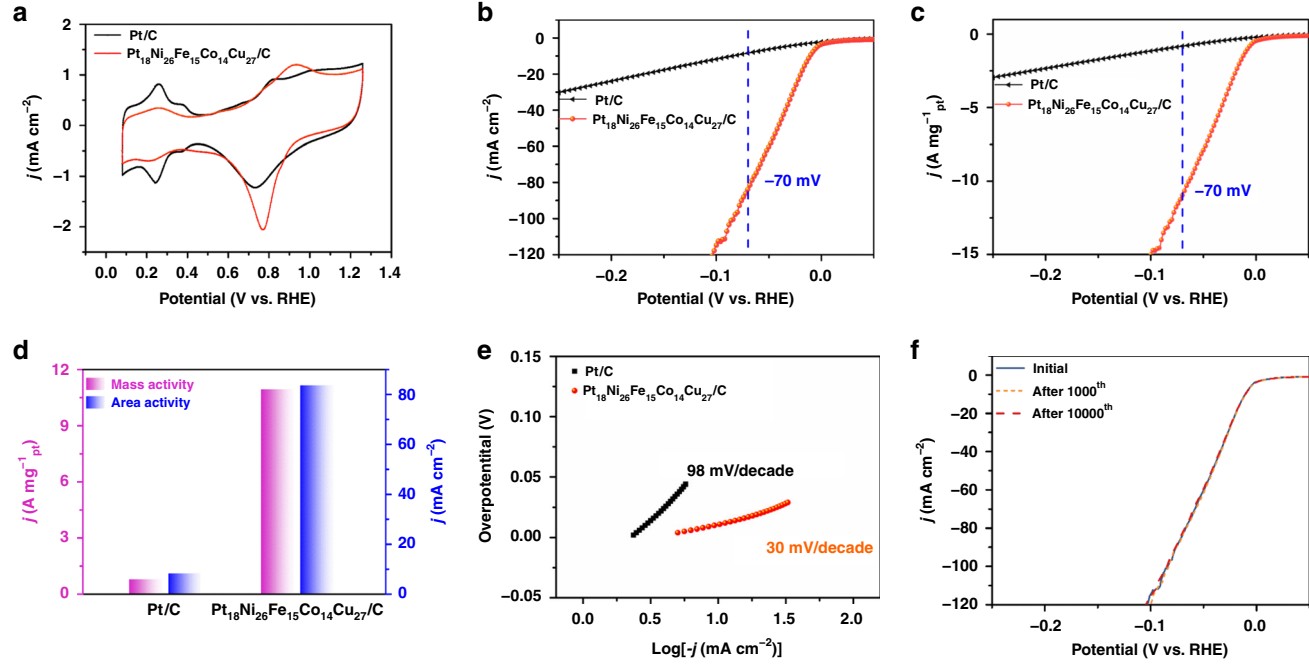

**Fig. 2 Electrocatalytic performance of the $Pt_{18}Ni_{26}Fe_{15}Co_{14}Cu_{27}$/C and Pt/C for HER in 1 M KOH electrolyte. a** CV curves. **b** HER polarization curves (geometrical area). **c** Pt mass loading normalized (mass activity) LSV curves. **d** Comparison of area activity and mass activity values for HER at −70 mV vs. RHE. **e** Tafel slope. **f** HER polarization curves (geometrical area) for $Pt_{18}Ni_{26}Fe_{15}Co_{14}Cu_{27}$/C with different CV cycle.

To better understand the HER performance of the $Pt_{18}Ni_{26}Fe_{15}$-$Co_{14}Cu_{27}$/C catalyst, we tested the electrochemical double-layer capacitance, electrochemical impedance spectra, and turnover frequency (TOF). We found that the electrochemical double-layer capacitance of $Pt_{18}Ni_{26}Fe_{15}Co_{14}Cu_{27}$/C catalyst is higher than that of commercial Pt/C catalyst (Supplementary Fig. 5), indicating that the

NPs-HEAs can expose more active sites. Also, electrochemical impedance spectra (Supplementary Fig. 6) exhibits $Pt_{18}Ni_{26}Fe_{15}$-$Co_{14}Cu_{27}$/C catalyst has a smaller semicircular diameter than commercial Pt/C catalyst, the transfer resistance of $Pt_{18}Ni_{26}Fe_{15}$-$Co_{14}Cu_{27}$/C is much lower than that of commercial Pt/C catalysts, indicating higher interfacial charge transfer rate and faster HER

kinetics. The TOF value is used to characterize the activity of each site in the catalyst[37]. It is found that the TOF value of the $Pt_{18}Ni_{26}Fe_{15}Co_{14}Cu_{27}$/C catalyst is higher than Pt/C catalyst under various potentials (Supplementary Fig. 7), which shows the faster HER kinetics of $Pt_{18}Ni_{26}Fe_{15}Co_{14}Cu_{27}$/C catalyst. These results further indicate that the NPs-HEAs can improve the catalytic activity effectively.

The stability of the catalysts was checked using the chronoamperometric method and 10,000th CV in 1.0 M KOH solution. The stability of the $Pt_{18}Ni_{26}Fe_{15}Co_{14}Cu_{27}$/C and Pt/C catalysts were measured at an overpotential of 11 mV and 84 mV (under current density 10 mA cm$^{-2}$ condition, Supplementary Fig. 8), respectively. After 10 h test, 99% of the current density is maintained for the $Pt_{18}Ni_{26}Fe_{15}Co_{14}Cu_{27}$/C catalyst, while the current density of the Pt/C catalyst is only kept about 53%. To further evaluate the stability of $Pt_{18}Ni_{26}Fe_{15}Co_{14}Cu_{27}$/C and Pt/C catalysts, the 1000th and 10,000th CV cycles in 1 M KOH solution were performed. Figure 2f displays the LSV curves of $Pt_{18}Ni_{26}Fe_{15}Co_{14}Cu_{27}$/C catalyst before and after 1000th and 10,000th CV cycles. There is no obvious negative shift at the current density of 10 mA cm$^{-2}$, exhibiting higher stability than Pt/C catalyst (Supplementary Fig. 9, a negative shift of ~33 mV at 10 mA cm$^{-2}$ after 10,000th CV cycles). In addition, the morphology (Supplementary Fig. 10), the atomic ratio (Supplementary Table 1), lattice spacing and the XRD peaks (Supplementary Fig. 11) of catalyst do not change significantly after the stability test. In addition, the slight variations in the stoichiometry (Supplementary Table 2) after the electrocatalytic test was caused by the leaching of non-noble metals (Ni, Fe, Co, and Cu) in the catalyst[38,39]. XPS analysis (Supplementary Fig. 12) shows that the Pt 4 f, Ni 2p, Fe 2p, Co 2p, and Cu 2p spectra of $Pt_{18}Ni_{26}Fe_{15}Co_{14}Cu_{27}$/C were very similar before and after electrocatalysis, while the oxide species increased slightly (Supplementary Table 3). However, the Pt/C catalyst shows obvious aggregation after the stability test (Supplementary Fig. 13). These results further suggest the excellent stability of the $Pt_{18}Ni_{26}Fe_{15}Co_{14}Cu_{27}$/C catalyst.

Furthermore, taking the change of Cu element content as an example, the effect of stoichiometry variations on the electrocatalytic performance is discussed. Specifically, this synthesis method is used to prepare PtNiFeCoCu NPs with other Cu content (PtNiFeCoCu$_{26}$ with slight stoichiometry variations, $Pt_{21}Ni_{27}Fe_{19}Co_{17}Cu_{16}$ and $Pt_{15}Ni_{24}Fe_{13}Co_{13}Cu_{35}$) by changing the amount of Cu precursor. From Supplementary Fig. 14, the morphology and size of the PtNiFeCoCu$_{26}$, $Pt_{21}Ni_{27}Fe_{19}Co_{17}Cu_{16}$ and $Pt_{15}Ni_{24}Fe_{13}Co_{13}Cu_{35}$ are similar to $Pt_{18}Ni_{26}Fe_{15}Co_{14}Cu_{27}$ NPs. The PXRD pattern (Supplementary Fig. 15) suggests the fcc structure of the PtNiFeCoCu$_{26}$, $Pt_{21}Ni_{27}Fe_{19}Co_{17}Cu_{16}$ and $Pt_{15}Ni_{24}Fe_{13}Co_{13}Cu_{35}$ NPs. The HER performance of PtNiFeCoCu$_{26}$/C catalyst with slight stoichiometry variations (Supplementary Figs. 16–24 and Supplementary Table 5) has no significant change compared with $Pt_{18}Ni_{26}Fe_{15}Co_{14}Cu_{27}$/C (PtNiFeCoCu$_{27}$/C). After activation (Supplementary Fig. 16), the PtNiFeCoCu$_{26}$/C catalyst has close overpotential (11 mV, Supplementary Fig. 17a), area activity and mass activity (82.66 mA cm$^{-2}$ and 10.81 A mg$^{-1}$$_{Pt}$ at −0.07 V vs. RHE, Supplementary Fig. 17a–c), Tafel slope (30 mV dec$^{-1}$, Supplementary Fig. 18a), electrochemical double-layer capacitance (7.01 mF, Supplementary Fig. 19a, d), electrochemical impedance (Supplementary Fig. 20a), TOF value (Supplementary Fig. 21a, c) and stability (Supplementary Figs. 22–24) compared with PtNiFeCoCu$_{27}$/C catalyst. In addition, the HER performance of $Pt_{21}Ni_{27}Fe_{19}Co_{17}Cu_{16}$/C and $Pt_{15}Ni_{24}Fe_{13}Co_{13}Cu_{35}$/C catalysts (Supplementary Figs. 16–24 and Supplementary Table 5) is slightly lower than the $Pt_{18}Ni_{26}Fe_{15}Co_{14}Cu_{27}$/C catalyst. The order of activity is as follows: $Pt_{18}Ni_{26}Fe_{15}Co_{14}Cu_{27}$/C (83.78 mA cm$^{-2}$ and 10.96 A

mg$^{-1}$$_{Pt}$ at −0.07 V vs. RHE) ≈ PtNiFeCoCu$_{26}$/C (82.66 mA cm$^{-2}$ and 10.81 A mg$^{-1}$$_{Pt}$ at −0.07 V vs. RHE) > $Pt_{21}Ni_{27}Fe_{19}Co_{17}Cu_{16}$/C (74.21 mA cm$^{-2}$ and 9.65 A mg$^{-1}$$_{Pt}$ at −0.07 V vs. RHE) > $Pt_{15}Ni_{24}Fe_{13}Co_{13}Cu_{35}$/C (59.76 mA cm$^{-2}$ and 7.88 A mg$^{-1}$$_{Pt}$ at −0.07 V vs. RHE).

**Electrocatalytic performance tests toward MOR.** To further explore the $Pt_{18}Ni_{26}Fe_{15}Co_{14}Cu_{27}$/C catalyst as redox bi-function electrocatalysts, we tested the MOR (oxidation reaction) activity. As shown in Fig. 3a and Supplementary Fig. 25, the $Pt_{18}Ni_{26}$-$Fe_{15}Co_{14}Cu_{27}$/C catalyst shows higher activity compared to Pt/C catalyst for MOR in 1 M KOH + 1 M $CH_3OH$ electrolyte at a sweep rate of 20 mV s$^{-1}$. And an onset potential (the mass activity of 0.1 A mg$^{-1}$$_{pt}$) was observed to decrease by 133 mV in the $Pt_{18}Ni_{26}Fe_{15}Co_{14}Cu_{27}$/C catalyst compared to the Pt/C catalyst (the inset of Fig. 3a), indicating that the activation barrier of methanol oxidation is lower. The $Pt_{18}Ni_{26}Fe_{15}Co_{14}Cu_{27}$/C catalyst achieves 10 times (4 times) higher in mass activity and area activity (15.04 A mg$^{-1}$$_{pt}$, 114.93 mA cm$^{-2}$) than that of Pt/C catalyst (1.45 A mg$^{-1}$$_{pt}$, 27.48 mA cm$^{-2}$) at peak potential for MOR (Fig. 3b). In addition, the MOR performance of PtNiFeCoCu$_{26}$/C catalyst is similar to that of the $Pt_{18}Ni_{26}Fe_{15}Co_{14}Cu_{27}$/C catalyst, the MOR performance of $Pt_{21}Ni_{27}Fe_{19}Co_{17}Cu_{16}$/C and $Pt_{15}Ni_{24}Fe_{13}Co_{13}Cu_{35}$/C catalysts is slightly lower than the $Pt_{18}Ni_{26}Fe_{15}Co_{14}Cu_{27}$/C catalyst. (Supplementary Fig. 26 and Supplementary Table 6). Moreover, among recently reported the Pt-based materials for MOR in alkaline medium, the $Pt_{18}Ni_{26}$-$Fe_{15}Co_{14}Cu_{27}$/C catalyst exhibits higher mass activity (Supplementary Table 8).

To study the stability of the $Pt_{18}Ni_{26}Fe_{15}Co_{14}Cu_{27}$/C and Pt/C catalysts in MOR, the chronoamperometry test at 0.65 V vs. RHE and 1000 CV cycles were executed. After 5000 s chronoamperometry test (Fig. 3c), the $Pt_{18}Ni_{26}Fe_{15}Co_{14}Cu_{27}$/C (3.79 A mg$^{-1}$$_{pt}$) catalyst shows higher stability than that of commercial Pt/C (0.20 A mg$^{-1}$$_{pt}$). After 1000 CV cycles, the mass activity of the $Pt_{18}Ni_{26}Fe_{15}Co_{14}Cu_{27}$/C and Pt/C catalysts decay by about 6.4% and 26.9% (Fig. 3d and Supplementary Fig. 28a), further confirmed the excellent stability of $Pt_{18}Ni_{26}Fe_{15}Co_{14}Cu_{27}$/C catalyst. From Supplementary Figs. 27, 28, the order of catalysts stability is as follows: $Pt_{18}Ni_{26}Fe_{15}Co_{14}Cu_{27}$/C ≈ PtNiFeCoCu$_{26}$/C > $Pt_{21}Ni_{27}Fe_{19}Co_{17}Cu_{16}$/C > $Pt_{15}Ni_{24}Fe_{13}Co_{13}Cu_{35}$/C. And the morphology and element ratio of PtNiFeCoCu/C catalyst with different rations did not obvious change after stability test (Supplementary Figs. 29, 30, Supplementary Table 1). However, the Pt/C catalyst shows obvious aggregation after the stability test (Supplementary Fig. 30d). XPS spectrum (Supplementary Fig. 31) shows that the Pt 4 f, Ni 2p, Fe 2p, Co 2p, and Cu 2p spectra of $Pt_{18}Ni_{26}Fe_{15}Co_{14}Cu_{27}$/C were very similar before and after MOR electrocatalysis. These test results indicate that the NPs-HEAs catalyst achieves double enhancement of catalytic activity and stability.

In MOR, the main route for catalyst deactivation is the poisoning effect of CO intermediates[40–42]. The $I_f/I_b$ ($I_f$: forward current density, $I_b$: backward current density) ratio of $Pt_{18}Ni_{26}$-$Fe_{15}Co_{14}Cu_{27}$/C catalyst (3.26) is larger than that of Pt/C (2.31), which shows the strong CO anti-poisoning performance of $Pt_{18}Ni_{26}Fe_{15}Co_{14}Cu_{27}$/C catalyst. And from CO stripping curves (Supplementary Fig. 32), the onset potentials of the $Pt_{18}Ni_{26}$-$Fe_{15}Co_{14}Cu_{27}$/C (0.391 V vs. RHE) display a 181 mV decrease compared with Pt/C (0.572 V vs. RHE) catalyst. PtNiFeCoCu$_{26}$/C (0.398 V vs. RHE), $Pt_{21}Ni_{27}Fe_{19}Co_{17}Cu_{16}$/C (0.395 V vs. RHE), and $Pt_{15}Ni_{24}Fe_{13}Co_{13}Cu_{35}$/C (0.413 V vs. RHE) catalysts have analogous onset potentials to $Pt_{18}Ni_{26}Fe_{15}Co_{14}Cu_{27}$/C. It further shows that the PtNiFeCoCu/C catalyst has better CO anti-poisoning performance.

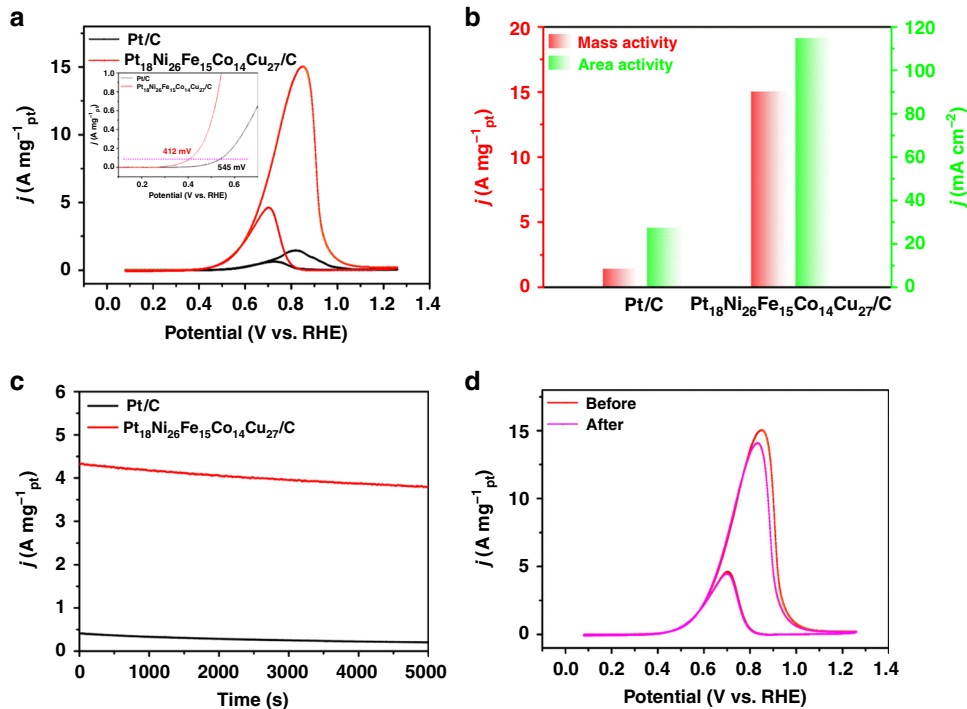

**Fig. 3 Methanol electro-oxidation performance of the Pt$_{18}$Ni$_{26}$Fe$_{15}$Co$_{14}$Cu$_{27}$/C and Pt/C in 1 M KOH + 1 M CH$_3$OH electrolyte. a** CV curves (the inset is the onset potential, the mass activity of 0.1 A mg$^{-1}_{pt}$) of Pt/C and Pt$_{18}$Ni$_{26}$Fe$_{15}$Co$_{14}$Cu$_{27}$/C. **b** Peak values of mass activity and area activity. **c** Chronoamperometric tests for MOR at 0.65 V vs. RHE. **d** CV curves of the Pt$_{18}$Ni$_{26}$Fe$_{15}$Co$_{14}$Cu$_{27}$/C before and after 1000 cycles.

**DFT studies**. We applied periodic DFT calculations to explore the HER and MOR performances in HEA. It is well known that the surface composition of the catalyst affects the catalytic performance. We use Ar$^+$ sputtering to study the change in surface element content of the PtNiFeCoCu NPs[43]. After Ar$^+$ sputtering for 10 s, the relative content of the elements has been reduced, but the reduced contents of Ni, Cu, and Fe were slightly higher than that of Pt and Co (Supplementary Table 4). We have also compared the PDOS of the HEA with slightly different stoichiometry. As shown in Supplementary Fig. 33, the comparison models show the highly similar electronic structure with the models applied in the work, in which the peak positions and patterns of d orbitals in each element display very limited change. Therefore, the slight stoichiometry variations of the HEA will not significantly affect the electronic structure of the PDOS results. Based on this, after a detailed comparison of different random atomic arrangements, the HEA structure with slightly Ni and Cu enriched surface has been applied as the lattice model due to the highest stability (Fig. 4a). From the side view, the lattice has shown a highly stable structure, in which the lattice shows subtle distortion after relaxation, which indicates good durability for electrocatalysis. Meanwhile, the surface Ni and Co dominate the electroactive region near the Fermi level ($E_F$) (Fig. 4b). To further understand the electronic structures, the partial projected density of states (PDOSs) of each element in HEA has been illustrated. Notably, Pt-5d occupies the deepest position near $E_V$ −4.5 eV (EV = 0 eV), playing as the electron reservoir for the reduction process such as HER. Both Co-3d and Ni-3d orbitals dominate the bands near $E_F$, which locate at $E_V$ −1.0 eV, contributing to the electron depletion center for HER and MOR. Moreover, the 3d orbitals of Cu, Co, and Fe not only alleviate the energy barrier of dual-way electron transfer for the oxidation and reduction process but also facilitate the stabilization of intermediates for MOR (Fig. 4c). A detailed study of the site-dependent PDOSs of each element was illustrated (Fig. 4d). Notably, only the surface Pt demonstrates an

evident upshift towards the $E_F$, which promotes the electron transfer for the HEA surfaces. Fe shows a site-independent electronic structure within HEA, which preserves the stable adsorption of intermediates with stronger anti-poisoning capability for the MOR process. From the bulk structure to the surface site, Co sites display an alleviation of the e$_g$-t$_{2g}$ splitting effect, supporting an enhanced electron transfer efficiency for electrocatalysis. Both Ni exhibits the relatively stable d-band-center to maintain the electroactive electron boosting center. For Cu sites closer to the surface, the 3d orbitals show a slight upshift trend to support the electroactivity of the HEA. Thus, the synergistic effect of multi-active sites on the HEA surface determines the remarkable performance of HER and MOR (Fig. 4d). For the HER process, the initial adsorption of water determines the efficiency of water-splitting and the following proton transfer. For the water adsorption process, the evident downshift of s,p orbitals in H$_2$O has been noticed, which confirms the active electron transfer from the HEA to the water to achieve the stable adsorption and lays a good foundation for the following water dissociation (Fig. 4e). Similarly, we notice the adsorption of CH$_3$OH on HEA indicates the evident downshifting of s,p bands, and overlapping with electroactive d orbitals of the surface (Fig. 4f). For the multi-electron involved MOR process, the linear correlation of intermediates transformation is the key to guarantee the proton and electron transfer. Such a linear correlation is noticed for the s,p orbitals of key intermediates along the MOR process, which not only supports the efficient oxidation of the intermediates but also leads to the optimal binding strength during the intermediate transformation. Thus, superior MOR performance is guaranteed in the HEA (Fig. 4g).

Then, we further interpret the reaction trend for both HER and MOR from both the structural configuration and energetic reaction pathways. The most stable structural configurations of four key initial reactants and intermediates have been displayed.

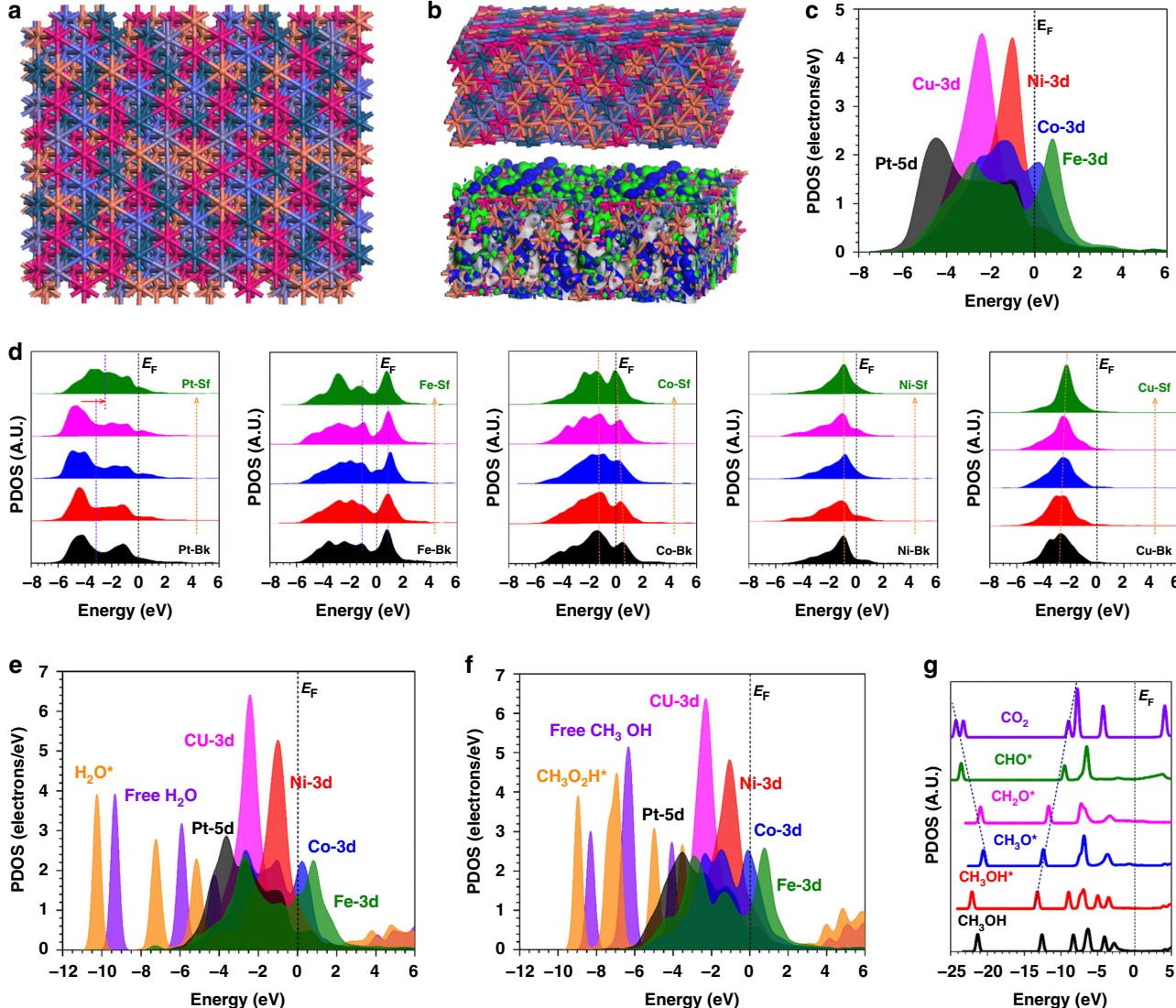

**Fig. 4 Density functional theory calculations for the structural configuration and PDOSs. a** The side view of the structural configuration of HEA. **b** The side view of HEA structural configuration and the real spatial contour plots for bonding and anti-bonding orbitals near $E_F$. And the top view of the real spatial contour plots for bonding and anti-bonding orbitals near $E_F$ for the HEA. Dark green balls = Pt; Gray balls= Fe; Blue balls = Co; Pink balls = Ni and Orange balls = Cu. **c** The PDOSs of the HEA. **d** The site-dependent PDOSs of Pt, Fe, Co, Ni, and Cu in HEA. **e** The PDOSs for the water adsorption. **f** The PDOSs for the $CH_3OH$ adsorption. **g** The PDOSs for the key intermediates of MOR.

The most stable adsorption of $CH_3OH$ and $H_2O$ locates near Ni and Fe sites, respectively. The OH is stabilized in the neighboring hollow sites, which avoids the active site blocking during both HER and MOR. Meanwhile, the H adsorption prefers the hollow site near Ni and Co, which is distinct from the OH. Thus, the multi-site adsorption for HER and MOR process on HEA guarantees superior performances (Fig. 5a). Moreover, we have supplied the adsorption sites mapping of the surface to support the HER mechanism (Fig. 5b). Notably, the different sites in HEA surfaces demonstrated very varied adsorption preference for the intermediates. For HER, the initial adsorption of $H_2O$ locates on the Fe sites, which activates the dissociation of water molecules and facilitates the stabilization of *OH in the neighboring hollow sites. Meanwhile, the nearby Ni, Co show the relatively preferred H* adsorption after the water dissociation, leading to the stabilization of H in the hollow sites surrounded by Co and Ni. The generated $H_2$ shows overall weak binding to the surface, indicating the quick desorption process to guarantee the efficient HER process. Therefore, our additional information has supplied

the binding stability of the intermediates for the HER process, which supports the Volmer–Heyrovsky mechanism. The HER process supports a continuous downhill trend, confirming the efficient proton and electron transfer (Fig. 5c). Owing to the multi-active sites for OH and H, the water dissociation demonstrates a low activation barrier of 0.11 eV for the transition state (Fig. 5d). For the MOR process, the rate-determining step occurs at [CHO* + 3*OH + 3H$_2$O] to [HCOOH + 2*OH + 4H$_2$O] with the largest energy barrier of 0.45 eV. The transition state displays the activation energy of 0.64 eV. The overall MOR process is exothermic, which releases 2.34 eV energy (Fig. 5e). The anti-poisoning capability is another essential requirement for the long-term application of MOR electrocatalysts. Compared to MOR, the formation of CO shows a much larger energy barrier of 0.81 eV and activation energy (0.94 eV), resulting in the suppression of the CO poisoning. The holistic reaction trend of the CO poisoning mechanism is also much weaker than the MOR process, which explains both the superior electroactivity and durability of the HEA (Fig. 5f).

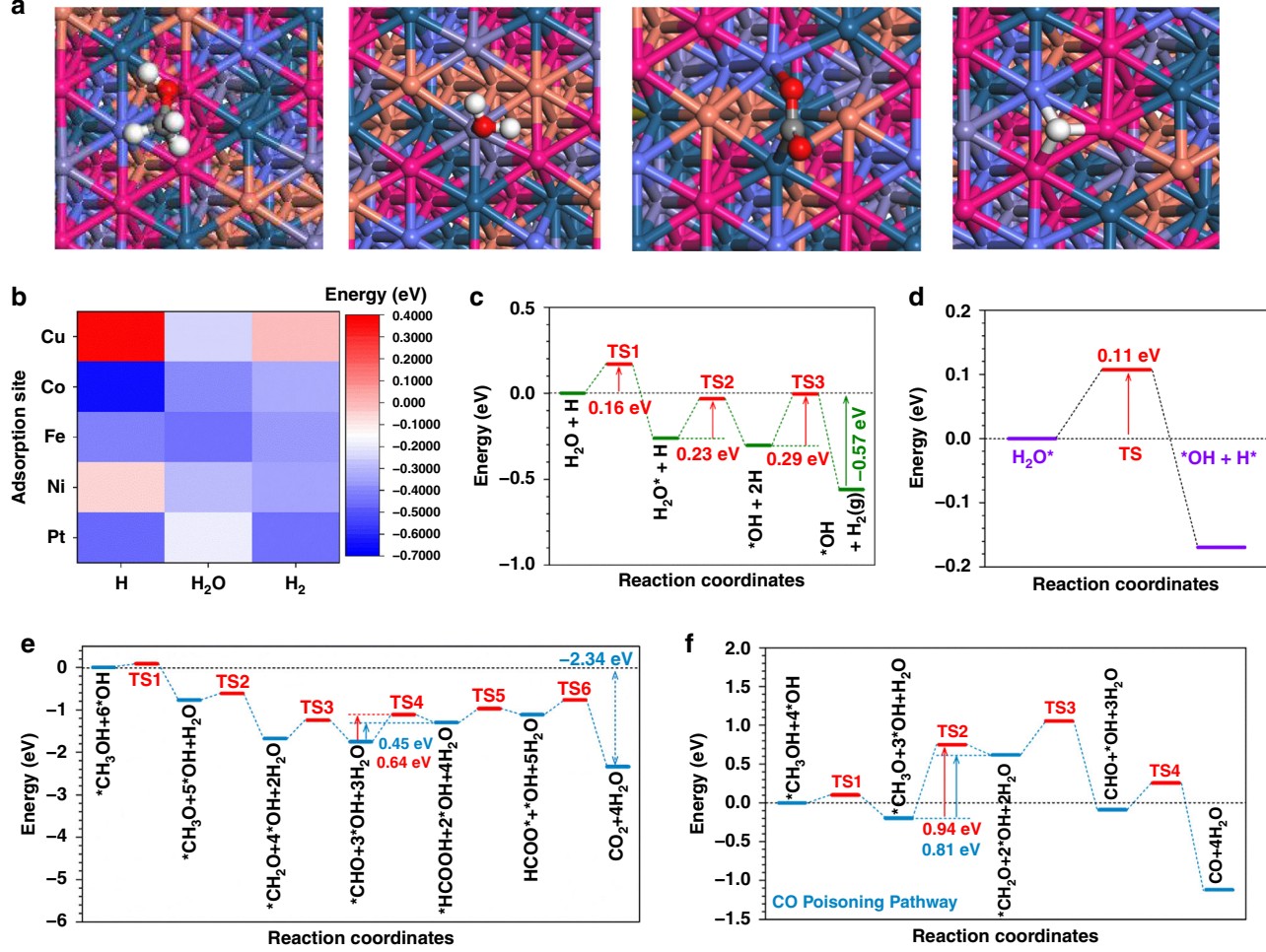

**Fig. 5 Density functional theory calculations for the structural configuration and energetic reaction pathways. a** The structural configuration of stable adsorption of key intermediates in HER and MOR. From left to right: CH₃OH, H₂O, CO₂, H. Dark green balls = Pt; Gray balls = Fe; Blue balls = Co; Pink balls = Ni; Orange balls = Cu; Red balls = O and White balls = H. **b** The binding energy mapping of HER. **c** The energetic pathway of the alkaline HER. **d** The activation energies of water dissociation. **e** The energetic pathway of the alkaline MOR. **f** The energetic pathway of CO poisoning.

## Discussion

In summary, we have synthesized the uniform and small size Pt₁₈Ni₂₆Fe₁₅Co₁₄Cu₂₇ NPs HEA nanoparticles by a simple low-temperature synthesis method at atmospheric pressure. Electro-catalytic test results showed that the obtained Pt₁₈Ni₂₆Fe₁₅-Co₁₄Cu₂₇/C catalyst has excellent bi-functional electrocatalytic properties for reduction reaction (HER) and oxidation reaction (MOR). Pt₁₈Ni₂₆Fe₁₅Co₁₄Cu₂₇/C catalyst showed an ultrasmall overpotential (11 mV at 10 mA cm$^{-2}$) and superior activity (10.96 A mg$^{-1}$$_{Pt}$ at −0.07 V vs. RHE) and stability for HER, which is one of the best HER activity in alkaline medium. And it is also the effective catalysts for MOR and displayed the excellent activity (15.04 A mg$^{-1}$$_{Pt}$) and better CO anti-poisoning in alkaline solution, which is the best alkaline MOR activity among the ever reported. Through DFT calculations, the origin of remarkable electroactivity and durability of HEA in HER and MOR is attributed to the synergistic effect of each element for efficient electron transfer. The suitable electronic environment of the HEA realizes the multi-active sites for appropriate adsorption of key intermediates and efficient electron transfer during the electrocatalysis, which maximizes the utilization of surface electroactivity. The simple oil phase synthesis strategy proposed in this work, as well as the multi-active sites and fast site-to-site electron transfer mechanism, which is expected to lay the foundation for the preparation of other HEAs and their application in related electrocatalysis.

## Methods

**Materials.** Platinum (II) acetylacetonate (Pt(acac)₂, 97%), nickel (II) acetylacetonate (Ni(acac)₂, 95%), Molybdenumhexacarbonyl (Mo(CO)₆, 98%), oleylamine (OAm, >70%) and glucose were bought from Sigma-Aldrich. Cupric acetylacetonate (Cu(acac)₂, 98%) was purchased from Energy Chemical. (1-Hexadecyl) trimethylammonium chloride (CTAC, 96%), Tris(2,4-pentanediaonato) Cobalt (III) (Co(acac)₃, 98%), Iron (III) 2,4-pentanedionate (Fe(acac)₃), Nafion solution (5 wt.%) were supplied by Alfa Aesar. Methanol, cyclohexane, ethanol, and isopropanol were bought from Beijing Tongguang Fine Chemicals Company. Potassium hydroxide (KOH, 90%) was purchased from Aladdin.

**Preparation of HEA Pt₁₈Ni₂₆Fe₁₅Co₁₄Cu₂₇ NPs.** CTAC (50 mg) was added into oleylamine (5 mL) in a 15 mL vial. After sonication for about 15 min, Pt(acac)₂ (10 mg), Ni(acac)₂ (6.4 mg), Fe(acac)₃ (8.8 mg), Co(acac)₃ (8.9 mg), Cu(acac)₂ (6.5 mg), glucose (60 mg), and Mo(CO)₆ (33 mg) were added into the vial. In order to obtain a homogeneous solution, the mixture was sonicated for 1 h. The vial was heated to 220 °C at 5 °C min$^{-1}$ and then kept 2 h under magnetic stirring at 400 rpm. The black colloidal products were collected by centrifugation and washed two times with an ethanol/cyclohexane mixture. Finally, the black colloidal products were kept in cyclohexane for further use.

**Preparation of alloy PtNiFeCoCu with different element ratios NPs.** CTAC (50 mg) was added into oleylamine (5 mL) in a 15 mL vial. After sonication for about 15 min, Pt(acac)₂ (10 mg), Ni(acac)₂ (6.4 mg), Fe(acac)₃ (8.8 mg), Co(acac)₃

(8.9 mg), Cu(acac)$_2$ (6.2 mg), glucose (60 mg) and Mo(CO)$_6$ (33 mg) were added into the vial. In order to obtain a homogeneous solution, the mixture was sonicated for 1 h. The vial was heated to 220 ºC at 5 ºC min$^{-1}$ and then kept 2 h under magnetic stirring at 400 rpm. The black colloidal products (PtNiFeCoCu$_{26}$) were collected by centrifugation and washed two times with an ethanol/cyclohexane mixture. Finally, the black colloidal products were kept in cyclohexane for further use. The synthesis of NPs with other element ratios only changes the amount of Cu (acac)$_2$ precursor (4.0 mg and 8.5 mg).

**Preparation of PtNiFeCoCu/C**. The obtained 1 mg HEA NPs dispersed in 10 mL cyclohexane was mixed with 4 mg of carbon (Ketjen Black-300) in 10 mL ethanol under sonication for 1 h, and then the product was collected via centrifugation with ethanol. The PtNiFeCoCu/C catalysts were further cleaned (remove organic species) with 0.5 M acetic acid (ethanol solution) under N$_2$ atmosphere. After being sonicating for 2 h, the products were collected by centrifugation and washed with ethanol for three times.

**Characterization**. The TEM and high-resolution TEM (HRTEM) images of the samples were characterized by an FEI Tecnai-G2 F30 at an accelerating voltage of 300 KV. Powder X-ray diffraction (XRD) spectra were recorded on X'Pert-PRO MPD diffractometer operating at 40 kV and 40 mA with Cu Kα radiation. The compositions of the HEA NPs were determined by the inductively coupled plasma atomic emission spectrometer (ICP-AES, Varian 710-ES). The catalysts after the durability tests were scratched off the glassy carbon electrode with the aid of sonication in ethanol and then collected for further TEM, XRD, and ICP characterization. The XPS analyses were carried out with Axis Supra spectrometer using a monochromatic Al Kα source (15 mA, 14 kV). Survey scan analyses were carried out with an analysis area of 300 × 700 μm$^2$ and a pass energy of 100 eV. High-resolution analyses were carried out with an analysis area of 300 × 700 μm$^2$ and a pass energy of 30 eV. Spectra have been corrected to the main line of the carbon 1s spectrum (adventitious carbon) set to 284.8 eV. Spectra were analyzed using CasaXPS software (version 2.3.14).

**Electrochemical measurements**. The 1 mg PtNiFeCoCu/C catalysts were dispersed in a mixture of 495 μL ultrapure water, 495 μL isopropanol, and 10 μL Nafion solution, after sonication for 1 h, PtNiFeCoCu/C catalyst with the concentration of 1 mg mL$^{-1}$ was obtained.

Electrochemical measurements were conducted on a CHI 760E Electrochemical Workstation (Shanghai Chenhua Instrument Corporation, China) in a conventional three-electrode cell. The graphite rod electrode as the counter electrode and a saturated calomel electrode (SCE) as the reference electrode. The working electrode was a glassy carbon electrode (GCE, diameter: 3 mm, area: 0.07065 cm$^2$). Ten microliters of the catalyst were dropped onto the GCE surface for further electrochemical tests. All the potentials reported in this work were converted to the reversible hydrogen electrode (RHE). Cyclic voltammograms (CVs) were performed in N$_2$-saturated 1 M KOH solution from 0 to 1.2 V vs. RHE at a scan rate of 20 mV s$^{-1}$. Electrochemical impedance spectroscopy (EIS) measurements were measured at −50 mV vs. RHE in the frequency range from 10 kHz to 0.01 Hz in N$_2$-saturated 1 M KOH solution.

**Hydrogen evolution reaction (HER) measurements**. The HER performance of the catalysts was evaluated by linear sweep voltammetry (LSV) with a scan rate of 5 mV s$^{-1}$ in N$_2$-saturated 1 M KOH solution, and all polarization curves were 95% iR-corrected. The durability tests were performed in 1.0 M KOH solution using the chronoamperometric method. And 1000th/10000th CVs were also measured to evaluate the stability of catalysts. 20% Pt/C catalyst was also dropped on the GCE as a reference catalyst for electrochemical tests under the same conditions.

**TOF calculation**. The CVs curve was collected from 0 to 1.2 V vs. RHE in 0.5 M H$_2$SO$_4$ solution with a scan rate of 20 mV s$^{-1}$:

$$n = \frac{Q_H}{2*F}, \qquad (1)$$

$$\text{TOF}(S^{-1}) = \frac{I}{2*F*n}, \qquad (2)$$

where $n$ is the number of active sites, number 2 represents two electrons (produce one hydrogen molecule), $Q_H$ represents electron transfer quantity (the hydrogen desorption peak), $F$ is Faraday's constant (96485.3 C mol$^{-1}$), and $I$ (A) is the current measured at a specific potential during LSV measurement.

**Methanol oxidation reaction (MOR) measurements**. The CVs for MOR were conducted in N$_2$-saturated 1 M KOH + 1 M CH$_3$OH solution between 0.2–1.2 V vs. RHE with a scan rate of 20 mV s$^{-1}$. For the MOR stability tests, chronoamperometric tests were performed at a fixed potential of 0.65 V vs. RHE, and 1000th CVs were also performed to evaluate the stability of catalysts.

**CO stripping curves were carried out in 1 M KOH solution**. Before the tests, 1 M KOH solution was first deaerated with high-purity N$_2$. Then, CO was bubbled into the cell for 15 min while the potential of the working electrode was held at a constant potential of 0.1 V vs. RHE. Then N$_2$ was bubbled into the system for 15 min to remove CO gas. After that, CO stripping curves were recorded between 0 and 1.2 V vs. RHE at a scan rate of 20 mV s$^{-1}$.

**Calculation setup**. For all the calculations within this work, we have applied the DFT calculations within CASTEP code[44]. The GGA and PBE exchange-correlation functionals are selected for all the calculations[45,46]. The cutoff energy of plane-wave basis sets based on the ultrasoft pseudopotential has been set to 440 eV with the selection of the algorithm Broyden–Fletcher–Goldfarb-Shannon (BFGS) for all the geometry optimizations[47]. To specifically discuss the electrocatalysis on the surface, we have applied the HEA model with similar stoichiometry as experimental characterizations. The HEA model has been built, which consists of 126 atoms in total. The atomic arrangements of different elements are constructed randomly by following the same ratio as the experiments of Pt$_{18}$Ni$_{26}$Fe$_{15}$Co$_{14}$Cu$_{27}$. Based on the components of the HEA by experimental characterizations, Ni and Cu have the highest concentration, which is 26% and 27%, respectively. To determine the most possible preferred model, we have compared the total energy of the HEA model with different surface arrangements, in which the present applied model with Cu and Ni slight rich feature have been the most stable one. Therefore, we have applied the HEA surface model with Cu and Ni rich feature. The reaction energy has been considered based on the intermediate adsorptions on the surface[34]. The Monkhost–Pack reciprocal space integration was performed using coarse k-points with a mesh of 2 × 2 × 1[48], which was guided by the initial convergence test. With these settings, the overall total energy for each step is converged to less than 5.0 × 10$^{-5}$ eV per atom. The Hellmann-Feynman forces on the atom were converged to less than 0.001 eV/Å.

## Data availability

The data that support the findings of this study are available from the corresponding author upon reasonable request.

## Code availability

All code supporting the findings of this study are available from the corresponding author on request.

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

## Acknowledgements

This work was supported by the National Natural Science Foundation of China (21571112, 51572136, 51772162, 51802171), the Taishan Scholars Program, Natural Science Foundation of Shandong Province, China (ZR2018BB031), Open Fund of the Key Laboratory of Eco-chemical Engineering (Qingdao University of Science and Technology, No. KF1702), the Taishan Scholar Project of Shandong Province (tsqn201909123)

## Author contributions

L.W. and J.L. conceived and supervised the research. J.L. and H.L. designed the experiments. H.L. performed most of the experiments and data analysis. B.H. performed the DFT calculations and mechanistic analysis. Y.H. and H.Z. prepared the electrodes and helped with electrochemical measurements. W.Q. and D.Z. conducted and analyzed HRTEM micrographs and mapping images. Y.Y. performed and analyzed XRD and ICP measurements. S.L. and W.C. analyzed XPS measurements. All authors discussed the results and commented on the manuscript.

## Competing interests

The authors declare no competing interests.
