## [Peer Review File · Nature Communications]

REVIEWER COMMENTS

Reviewer #1 (Remarks to the Author):

The authors describe dual function high entropy alloy based electrocatalysts in a combined theoretical and experimental approach. The studied materials comprise nanoparticles containing Pt, Ni, Fe, Co, Cu in a fixed stoichiometry accessed via an oil phase synthesis. The catalysts are studied supported on C and show activity towards HER and the methanol oxidation reaction, whereas for both HER and MOR in alkaline medium, very good overpotentials, as well as mass activities are reported.

The materials are characterized by a suite of state-of-the-art methods, supplemented by DFT calculations to elucidate active site atomic and electronic structure. Overall, the study appears to be well-performed with data supporting the hypothesis made.

My requests concern mostly i) structural development of the materials (surface changes/ alloying) and ii) experimental design and conceptualization:

1) The stoichiometry of the HEA nanoparticles has been determined by ICP-OES and emerged during synthesis. Why has this particular composition been chosen? I can imagine that the selection of the right (= most optimal) stoichiometry in a 5 D parameter space is incredibly difficult. What is the rationale behind this particular choice?

2) Conceptually and from experiment design, it is unclear to me, why this particular composition (stoichiometry and choice of the 5 metals) has been investigated and has succeeded apparently so tremendously. I would like to see how sensitive HER or MOR performance responds to slight stoichiometry variations, i.e. how much better or worse is Pt₁₈Ni₂₆Fe₁₅Co₁₄Cu₂₆/C compared to the sample here Pt₁₈Ni₂₆Fe₁₅Co₁₄Cu₂₇/C? I feel that the manuscript could improve significantly and increase in impact by better motivating these choices and giving them a scientific basis. Now, it seems to be a matter of chance to find such a super-catalyst. How often has the synthesis been reproduced to obtain exactly the same composition of NPs?

3) Does the obtained stoichiometry form an optimum in reactivity or is it only a local maximum? Please discuss this in terms of predicting the most optimal alloy composition and possible synthetic stoichiometry control?

4) Does the stoichiometry stay constant under (prolonged) HER operation? Supplementary table points to that, but can the slight variations not be caused by leaching? Please check by preferably ICP-MS of the spent electrolyte.

5) The same sensitivity question I have regarding the electronic structure calculations: Did the authors check different random structures and slightly stoichiometry variations to check sensitivity of the PDOS distribution?

6) Although the study presents an interesting case of a new catalyst material, there should be more conceptual rationale presented to guide and motivate the choices w.r.t. elements and composition made in this manuscript. Now it is more like a post-factum explanation of the observations by DFT calculations.

7) The authors conclude with: "The optimized electronic environment constructs the strategy of the multi-active site for adsorption and transformation of key intermediates, which maximizes the utilization of surface electroactivity." What does "optimized" mean here? There has been neither design nor optimization involved in the present study – rather an arbitrary composition alloy has been presented here with remarkable activities.

8) technical details: XPS part: There are oxidized species reported. From the (incomplete) experimental details, it is unclear whether these oxidized metal species (Pt²⁺, Ni²⁺, Cu²⁺...) arise

from the synthesis or from sample handling in air. Please elaborate. XPS under inert atmosphere transfer should be presented. Also, XPS post catalysis should be presented to check for electronic structure changes/ oxidation of the materials. Furthermore, XPS experimental details should be updated as suggested in e.g. <http://www.xpsfitting.com/search/label/Publication%20Methods>

Reviewer #2 (Remarks to the Author):

This work reports the synthesis, characterization and reactivity towards electrocatalytic performance for hydrogen evolution reaction (HER) and methanol oxidation reaction (MOR) of Pt₁₈Ni₂₆Fe₁₅Co₁₄Cu₂₇/C NPs. Different techniques are used, experimental and theoretical. The methods used are state-of-the-art and well chosen. The paper claims to discovery of a particular efficient and selective electro-catalyst, which is to my opinion justified. Moreover, the excellent work is supported by pertinent results and will be of no doubt of great interest to others in the community. The research is in line with the investigation strategy based on multi-metal alloys (more than 3 metals). I have only one comment that the authors might consider in their future works: This very complex question is particularly well suited to be studied with the statistical machine learning simulation methods, and would be a perfect example to be tested. The chemical space (5 metals) is particularly challenging.

As far as the DFT calculations are concerned I want to say that the analysis of the electronic properties (PDOS, band gaps and orbital overlaps, ...) are very pedagogically presented and well interpreted, which is important for the readability of a wide public. The MS is well written and the graphs and figures are clear. I would like to congratulate the authors for their nice work. The work can be published after introducing some minor changes (see above and below).

Small comments:

I suggest the authors to precise the term "DFT" by completing it to "Periodic DFT".

Reviewer #3 (Remarks to the Author):

This work reports a new synthesis route for Pt-containing HEA nanoparticles which showed excellent HER and MOR activities and stability. The performance was explained with the help of DFT calculations which suggest synergistic effect from multiple surface sites for the two reactions. The work has potential to interest a broad field of readers. But a few issues need to be addressed before its acceptance for publication in Nat. Commun.

- In the DFT studies, the authors need to explain why Ni and Cu are slightly enriched on the surface. According to the XPS result of the fresh sample, it appears Pt should be enriched on the surface. Actually XPS spectra of the used samples after HER and MOR should be collected to analyze the surface composition change. It appears from the ICP analysis (Table S1) that Pt relative ratio increases after the two reactions. For more accurate analysis of the surface composition, XPS is not recommended unless energy-scanned XPS at a synchrotron source is conducted to get more surface sensitive information. Low energy ion scattering is more suited for top surface composition analysis.

- More experimental details should be provided for the synthesis of the HEAs and HEAs/C samples. Is there any surfactant left on the surface even after the activation before the electrocatalysis tests?

- Did the authors measure the Pt size on the Pt/C sample? I could not seem to find the information in the paper. The Pt size should be at least similar to that of the HEA particles so the electrocatalytic activity can be reasonably compared based on surface Pt sites.

- A minor note for table S1, the number of digit after the decimal should be consistent for all measurements: the fresh sample results have no decimals while the used sample results show 1 digit after the decimal. The same problem is present for tables S2 and S3.

-

Reply to Reviewers' Comments

Dear Reviewers,

Thank you for your precious time to constructive comments on our manuscript titled “**Fast Site-to-site Electron Transfer of High-entropy Alloy Nanocatalyst Driving Redox Electrocatalysis**” (Manuscript ID: NCOMMS-20-29577) for *Nature Communications*. We sincerely appreciate your opinions and confirmation of our work. Accordingly, we have supplied the corresponding response and revision based on the comments. We sincerely hope that our responses will fully address your concerns about our work.

To Reviewer 1:

General Comment: The authors describe dual function high entropy alloy based electrocatalysts in a combined theoretical and experimental approach. The studied materials comprise nanoparticles containing Pt, Ni, Fe, Co, Cu in a fixed stoichiometry accessed via an oil phase synthesis. The catalysts are studied supported on C and show activity towards HER and the methanol oxidation reaction, whereas for both HER and MOR in alkaline medium, very good overpotentials, as well as mass activities are reported. The materials are characterized by a suite of state-of-the-art methods, supplemented by DFT calculations to elucidate active site atomic and electronic structure. Overall, the study appears to be well-performed with data supporting the hypothesis made.

Author Reply: Thanks for your positive comments and support for our work. To further improve the quality of this manuscript as well as address your concerns, we have revised our manuscript as your suggestions. In addition, we also supplied a point-by-point response as follows. We wished the revised manuscript can fulfill your high requirements for the publication in *Nature Communications*.

My requests concern mostly i) structural development of the materials (surface changes/ alloying) and ii) experimental design and conceptualization:

Question 1: The stoichiometry of the HEA nanoparticles has been determined by ICP-OES and emerged during synthesis. Why has this particular composition been chosen? I can imagine that the selection of the right (= most optimal) stoichiometry in a 5 D parameter space is incredibly difficult. What is the rationale behind this particular choice?

Author Reply 1: Thanks very much for reviewer's comment. In the field of electrocatalysis, Pt-based alloys are effective catalysts for HER and MOR. However, reducing the Pt content while maintaining or even enhancing the overall electrocatalytic activity and durability remains a huge challenge. Considering

the high price and low earth abundance of Pt, there is a great need for multi-component Pt-based nanoalloys to reduce the content of Pt and further enhance the catalytic activity.

Nano high-entropy alloys, due to the endless combination of elements, provide unlimited possibilities for adjusting the electronic properties of the alloy and maximizing the catalytic activity. In addition, transition metal elements have also been extensively studied in the field of electrocatalysis. Taking into account the high abundance of Ni, Fe, Co, and Cu and they can easily form solid solutions with Pt. Here, Pt and four transition metal elements are controllably incorporated into a single nanostructure phase. [Small, 2019, 1904180. *Sci. China Mater.* 2018, 61, 2–22]

As described in the DFT calculation section of the revised manuscript, the role of each element in HER and MOR is pointed out, which provides a theoretical basis for material design. In order to uniformly mix these elements in a single-phase nanostructure, the Pt, Ni, Fe, Co, and Cu precursors with equal atomic weight were mixed together, but finally alloy nanoparticles with a molar ratio of Pt₁₈Ni₂₆Fe₁₅Co₁₄Cu₂₇ were obtained. And taking this catalyst as an example, the unique catalytic performance of HEA on HER and MOR was studied. As the reviewer pointed out, the selection of the right (= most optimal) stoichiometry in a 5 D parameter space is incredibly difficult, and the workload is huge. By changing the Cu content as an example, PtNiFeCoCu NPs with slight stoichiometry variations (PtNiFeCoCu₂₆) and other Cu contents (Pt₂₁Ni₂₇Fe₁₉Co₁₇Cu₁₆, Pt₁₅Ni₂₄Fe₁₃Co₁₃Cu₃₅) can be prepared using this synthetic approach by changing the amount of Cu precursors, and the influence on electrocatalytic performance were discussed and supplemented. Please refer to **Question 2** and **Question 3**.

Question 2: Conceptually and from experiment design, it is unclear to me, why this particular composition (stoichiometry and choice of the 5 metals) has been investigated and has succeeded apparently so tremendously. I would like to see how sensitive HER or MOR performance responds to slight stoichiometry variations, i.e. how much better or worse is Pt₁₈Ni₂₆Fe₁₅Co₁₄Cu₂₆/C compared to the sample here Pt₁₈Ni₂₆Fe₁₅Co₁₄Cu₂₇/C? I feel that the manuscript could improve significantly and increase in impact by better motivating these choices and giving them a scientific basis. Now, it seems to be a matter of chance to find such a super-catalyst. How often has the synthesis been reproduced to obtain exactly the same composition of NPs?

Author Reply 2: Thanks for reviewer's valuable comment and suggestion. In the early stage of DFT calculation, we compared several models with slightly different random structures. Among them, the current model used in the manuscript is the most stable model. In the calculations, we learned that the slight stoichiometry variations of the HEA will not significantly affect the electronic structure of the PDOS results (**Figure R1**). The comparison models show the highly similar electronic structure with the models applied in our work, in which the peak positions and patterns of d-orbitals in each element display very limited change. More importantly, the electronic structure of HEA is more related to the surrounding

elements and coordination environment. Therefore, the slight stoichiometry variations of the HEA will not significantly affect its electrocatalytic performance.

Figure R1. The PDOS comparison of the slight stoichiometry variations in HEA structures. (a) The PDOS of applied HEA model in our work. (b)-(d) The PDOS of the different HEA structures with slightly varied stoichiometry.

Following the reviewer’s suggestion, we prepared the alloy nanoparticles (PtNiFeCoCu_{26}) with slight stoichiometry variations by changing the amount of copper precursor and a series of tests were carried out. Experimental results show that the HER or MOR performance has not changed significantly.

In addition, during the experiment, by recording processing parameters, such as precursor concentration, reducing agent, ligand type/concentration, reaction temperature, heating rate, stirring rate, etc., try to keep the same synthesis conditions every time, the probability of getting the same composition of nanoparticles is still higher.

Figure R2. (a) The TEM image, (b) histogram of the diameter and (c) XRD pattern of PtNiFeCoCu_{26} NPs. (d) The TEM image of $\text{PtNiFeCoCu}_{26}/\text{C}$.

From the **Figure R2**, the PtNiFeCoCu₂₆ NPs size (about 3.4 nm, **Figure R2 a and b**) is consistent with the size of PtNiFeCoCu₂₇ (Pt₁₈Ni₂₆Fe₁₅Co₁₄Cu₂₇) particles (about 3.4 nm, **Figure 1a** in the revised manuscript). The powder X-ray diffraction (PXRD) pattern (**Figure R2c**) suggests the fcc structure of the PtNiFeCoCu₂₆ NPs and PtNiFeCoCu₂₇ NPs. After the NPs were supported on carbon, we can see that the PtNiFeCoCu₂₆ NPs were uniformly loaded on the carbon (**Figure R2d**).

Figure R3. Electrochemical performance of the PtNiFeCoCu₂₇/C and PtNiFeCoCu₂₆/C for HER in 1 M KOH electrolyte. (a) CV curves. (b) HER polarization curves (geometrical area) and (c) Pt mass loading normalized (mass activity) LSV curves. (d) Comparison of area activity and mass activity values for HER at -70 mV vs. RHE. (e) Tafel slope. (f) HER polarization curves (geometrical area) for PtNiFeCoCu₂₆/C after 10000 CV cycle.

Figure R3a shows the CV curves of PtNiFeCoCu₂₇/C and PtNiFeCoCu₂₆/C catalysts in N₂-saturated 1 M KOH at a scan rate of 20 mV s⁻¹. After activation, the linear sweep voltammetry (LSV) curve of the PtNiFeCoCu₂₆/C catalyst displays a low overpotential of 11 mV at the current density of 10 mA cm⁻² (normalized to the electrode area), which is consistent with the PtNiFeCoCu₂₇/C catalyst (**Figure R3b**). As shown in **Figure R3b** and **Figure R3d**, the area activity (normalized to the geometric area) of PtNiFeCoCu₂₆/C catalyst reaches 82.66 mA cm⁻² at -0.07 V vs. RHE, which is close to area activity of the PtNiFeCoCu₂₇/C catalyst (83.78 mA cm⁻² at -0.07 V vs. RHE). From the **Figure R3c** (LSV curves, normalized to the Pt mass), the mass activity for PtNiFeCoCu₂₆/C and PtNiFeCoCu₂₇/C catalyst are 10.81 A mg⁻¹Pt and 10.96 A mg⁻¹Pt at -0.07 V vs. RHE (**Figure 3d**), respectively. As shown in **Figure R3e**, the Tafel slopes of PtNiFeCoCu₂₆/C and PtNiFeCoCu₂₇/C are 30 mV dec⁻¹, which demonstrate that the HEAs catalyst greatly boosts HER kinetics. **Figure R3f** displays the LSV curves of PtNiFeCoCu₂₆/C catalyst before and after 10000th CV cycles. There is no obvious negative shift at the current density of 10 mA cm⁻², exhibiting the higher stability.

Figure R4. (a) CV curves in the double layer region at scan rates of 20, 40, 60, 80 and 100 mV s^{-1} in 1.0 M KOH and (b) Current density as a function of scan rate derived from (a) at 1.0 V (vs. RHE) of PtNiFeCoCu₂₆/C. (c) CV curves of in 0.5 M H₂SO₄ and (d) The potential dependent TOF curves of PtNiFeCoCu₂₇/C and PtNiFeCoCu₂₆/C.

The electrochemical double-layer capacitance of PtNiFeCoCu₂₆/C catalyst (7.01 mF, **Figure R4a and b**) is consistent with PtNiFeCoCu₂₇/C catalyst (7.00 mF, **Supplementary Figure 5**), indicating that the catalyst can expose more active sites. The TOF value is used to characterize the activity of each site in the catalyst. It is found that the TOF value of the PtNiFeCoCu₂₆/C catalyst is close to that of the PtNiFeCoCu₂₇/C catalyst under various potentials (**Figure R4d**), and it is slightly higher than that of the PtNiFeCoCu₂₇/C catalyst at some potentials, which shows the faster HER kinetics of catalyst.

Figure R5. (a) Chronoamperometric measurement curves and (b) The Nyquist plots at -50 mV vs. RHE of PtNiFeCoCu₂₇/C and PtNiFeCoCu₂₆/C.

The stability of the PtNiFeCoCu₂₇/C and PtNiFeCoCu₂₆/C catalysts were measured at an overpotential of 11 mV (under current density 10 mA cm⁻² condition, **Figure R5a**), respectively. After 10 h test, the current density has almost no decay. Also, electrochemical impedance spectra (**Figure R5b**) exhibits PtNiFeCoCu₂₆/C catalyst has a similar semicircular diameter with PtNiFeCoCu₂₇/C catalyst, indicating higher interfacial charge transfer rate and faster HER kinetics.

Figure R6. Methanol electro-oxidation performance of the PtNiFeCoCu₂₇/C and PtNiFeCoCu₂₆/C in 1 M KOH + 1 M CH₃OH electrolyte. (a) CV curves (area activity). (b) CV curves (mass activity). (c) Peak values of mass activity and area activity. (d) Chronoamperometric tests for MOR at 0.65 V vs. RHE. (e) CV curves of the PtNiFeCoCu₂₆/C before and after 1000 cycles. (f) The CO stripping curves for PtNiFeCoCu₂₆/C in 1 M KOH electrolyte.

As shown in **Figure R6a** and **b**, the PtNiFeCoCu₂₆/C catalyst shows similar activity compared to PtNiFeCoCu₂₇/C catalyst for MOR in 1 M KOH + 1 M CH₃OH electrolyte at a sweep rate of 20 mV s⁻¹. At peak potential for MOR (**Figure R6c**), the mass activity and area activity (14.96 A mg⁻¹_{pt}, 114.36 mA cm⁻²) of the PtNiFeCoCu₂₆/C catalyst are similar to those of the PtNiFeCoCu₂₇/C catalyst (15.04 A mg⁻¹_{pt}, 114.93 mA cm⁻²).

To study the stability of the PtNiFeCoCu₂₇/C and PtNiFeCoCu₂₆/C catalysts in MOR, the chronoamperometry test at 0.65 V vs. RHE and 1000 CV cycles were executed. After 5000 s chronoamperometry test (**Figure R6d**), the PtNiFeCoCu₂₇/C and PtNiFeCoCu₂₆/C catalysts all show high stability. After 1000 CV cycles, the mass activity of the PtNiFeCoCu₂₇/C and PtNiFeCoCu₂₆/C catalysts decay by about 6.4% and 5.2% (**Figure R6e** and **Figure 3d** in the revised manuscript), further confirmed the excellent stability of the catalysts.

In MOR, the main route for catalyst deactivation is the poisoning effect of CO intermediates. From CO stripping curves (**Figure R6f** and **Supplementary Figure 32**), the onset potentials of the PtNiFeCoCu₂₇/C

(0.391 V vs. RHE) only display 7 mV decrease compared with PtNiFeCoCu₂₆/C (0.398 V vs. RHE) catalyst. It further shows that the PtNiFeCoCu₂₇/C and PtNiFeCoCu₂₆/C catalysts have better CO anti-poisoning performance.

Figure R7. The TEM images of PtNiFeCoCu₂₆/C after stability test for (a) HER and (b) MOR.

From the **Figure R7**, the morphology of PtNiFeCoCu₂₆/C catalyst does not change significantly after the HER and MOR stability test.

Manuscript Revision: The corresponding sentence “We have also compared the PDOS of the HEA with slightly different stoichiometry. As shown in **Supplementary Figure 33**, the comparison models show the highly similar electronic structure with the models applied in the work, in which the peak positions and patterns of d-orbitals in each element display very limited change. Therefore, the slight stoichiometry variations of the HEA will not significantly affect the electronic structure of the PDOS results.” has also been added in the revised manuscript (please see **1-5 lines** (the red-label part) of **Page 11** in the revised manuscript). The **Figure R1** has been given in the Supporting Information (please see **Supplementary Figure 33**).

The related material preparation, characterization and catalytic performance discussion have been added to the revised manuscript. (please see **4-23 lines** (the red-label part) of **Page 8**, **1-2** and **12-15 lines** (the red-label part) of **Page 9**, **1-3** and **15-17 lines** (the red-label part) of **Page 10**, in the revised manuscript. **10-18 lines** (the red-label part) of **Page 2** in the Supporting Information). The **Figure R2-R7** has been given in the Supporting Information (please see **Supplementary Figure 14-24**, **Supplementary Figure 26-28**, **Supplementary Figure 30**, **Supplementary Figure 32**, **Supplementary Table 5** and **Supplementary Table 6**).

Question 3: Does the obtained stoichiometry form an optimum in reactivity or is it only a local maximum? Please discuss this in terms of predicting the most optimal alloy composition and possible synthetic stoichiometry control?

Author Reply 3: Thanks for reviewer's valuable comment and suggestion. Following the reviewer's suggestion, we discussed the synthetic stoichiometry control of the catalyst. It is very difficult to select the correct (=optimal) stoichiometric ratio in the 5D parameter space, and it is a lot of work to adjust the stoichiometric ratio. Here, by changing the Cu content as an example, PtNiFeCoCu NPs with other Cu contents ($\text{Pt}_{21}\text{Ni}_{27}\text{Fe}_{19}\text{Co}_{17}\text{Cu}_{16}$, $\text{Pt}_{15}\text{Ni}_{24}\text{Fe}_{13}\text{Co}_{13}\text{Cu}_{35}$ and PtNiFeCoCu_{26} mentioned in **Question 2**) can be prepared using this synthetic approach by changing the addition of Cu precursors. It can be seen from the experimental results that the $\text{Pt}_{21}\text{Ni}_{27}\text{Fe}_{19}\text{Co}_{17}\text{Cu}_{16}$ and $\text{Pt}_{15}\text{Ni}_{24}\text{Fe}_{13}\text{Co}_{13}\text{Cu}_{35}$ catalysts activity is slightly lower than the $\text{Pt}_{18}\text{Ni}_{26}\text{Fe}_{15}\text{Co}_{14}\text{Cu}_{27}/\text{C}$ catalyst.

Figure R8. (a) The TEM image and (c) histogram of the diameter of $\text{Pt}_{21}\text{Ni}_{27}\text{Fe}_{19}\text{Co}_{17}\text{Cu}_{16}$ NPs. (b) The TEM image and (d) histogram of the diameter of $\text{Pt}_{15}\text{Ni}_{24}\text{Fe}_{13}\text{Co}_{13}\text{Cu}_{35}$ NPs.

From the **Figure R8**, the $\text{Pt}_{21}\text{Ni}_{27}\text{Fe}_{19}\text{Co}_{17}\text{Cu}_{16}$ NPs and $\text{Pt}_{15}\text{Ni}_{24}\text{Fe}_{13}\text{Co}_{13}\text{Cu}_{35}$ NPs sizes (about 3.6 and 3.4 nm) are similar to the size of $\text{Pt}_{18}\text{Ni}_{26}\text{Fe}_{15}\text{Co}_{14}\text{Cu}_{27}$ particles (about 3.4 nm, **Figure 1a** in the revised manuscript).

Figure R9. (a) XRD patterns. (b) The TEM image of $\text{Pt}_{21}\text{Ni}_{27}\text{Fe}_{19}\text{Co}_{17}\text{Cu}_{16}/\text{C}$. (c) The TEM image of $\text{Pt}_{15}\text{Ni}_{24}\text{Fe}_{13}\text{Co}_{13}\text{Cu}_{35}/\text{C}$.

The powder X-ray diffraction (PXRD) patterns (**Figure R9a**) of the NPs products also reveal the typical fcc structure. After compounding with carbon, the $\text{Pt}_{21}\text{Ni}_{27}\text{Fe}_{19}\text{Co}_{17}\text{Cu}_{16}/\text{C}$ NPs and $\text{Pt}_{15}\text{Ni}_{24}\text{Fe}_{13}\text{Co}_{13}\text{Cu}_{35}$ were uniformly loaded on the carbon (**Figure R9b and c**).

Figure R10. Electrocatalytic performance of the $\text{Pt}_{21}\text{Ni}_{27}\text{Fe}_{19}\text{Co}_{17}\text{Cu}_{16}/\text{C}$, $\text{Pt}_{18}\text{Ni}_{26}\text{Fe}_{15}\text{Co}_{14}\text{Cu}_{27}/\text{C}$ and $\text{Pt}_{15}\text{Ni}_{24}\text{Fe}_{13}\text{Co}_{13}\text{Cu}_{35}/\text{C}$ for HER in 1 M KOH electrolyte. (a) CV curves. (b) HER polarization curves (geometrical area) and (c) Pt mass loading normalized (mass activity) LSV curves. (d) Comparison of area activity and mass activity values for HER at -70 mV vs. RHE. (e) Tafel slope. (f) Chronoamperometric measurement curves.

Figure R10a shows the CV curves of $\text{Pt}_{21}\text{Ni}_{27}\text{Fe}_{19}\text{Co}_{17}\text{Cu}_{16}/\text{C}$, $\text{Pt}_{18}\text{Ni}_{26}\text{Fe}_{15}\text{Co}_{14}\text{Cu}_{27}/\text{C}$ and $\text{Pt}_{15}\text{Ni}_{24}\text{Fe}_{13}\text{Co}_{13}\text{Cu}_{35}/\text{C}$ catalysts in N_2 -saturated 1 M KOH at a scan rate of 20 mV s^{-1} . After activation, the

linear sweep voltammetry (LSV) curve of $\text{Pt}_{21}\text{Ni}_{27}\text{Fe}_{19}\text{Co}_{17}\text{Cu}_{16}/\text{C}$, $\text{Pt}_{18}\text{Ni}_{26}\text{Fe}_{15}\text{Co}_{14}\text{Cu}_{27}/\text{C}$ and $\text{Pt}_{15}\text{Ni}_{24}\text{Fe}_{13}\text{Co}_{13}\text{Cu}_{35}/\text{C}$ catalysts display overpotential of 12 mV, 11 mV and 15 mV at the current density of 10 mA cm^{-2} (normalized to the electrode area, **Figure R10b**), respectively. As shown in **Figure R10b** and **Figure R10d**, the area activity (normalized to the geometric area) of $\text{Pt}_{21}\text{Ni}_{27}\text{Fe}_{19}\text{Co}_{17}\text{Cu}_{16}/\text{C}$, $\text{Pt}_{18}\text{Ni}_{26}\text{Fe}_{15}\text{Co}_{14}\text{Cu}_{27}/\text{C}$ and $\text{Pt}_{15}\text{Ni}_{24}\text{Fe}_{13}\text{Co}_{13}\text{Cu}_{35}/\text{C}$ catalysts reach 74.21 mA cm^{-2} , 83.78 mA cm^{-2} and 59.76 mA cm^{-2} at -0.07 V vs. RHE, respectively. From the **Figure R10c** (LSV curves, normalized to the Pt mass), the mass activity for $\text{Pt}_{21}\text{Ni}_{27}\text{Fe}_{19}\text{Co}_{17}\text{Cu}_{16}/\text{C}$, $\text{Pt}_{18}\text{Ni}_{26}\text{Fe}_{15}\text{Co}_{14}\text{Cu}_{27}/\text{C}$ and $\text{Pt}_{15}\text{Ni}_{24}\text{Fe}_{13}\text{Co}_{13}\text{Cu}_{35}/\text{C}$ catalysts are $9.65 \text{ A mg}^{-1}_{\text{Pt}}$, $10.96 \text{ A mg}^{-1}_{\text{Pt}}$ and $7.88 \text{ A mg}^{-1}_{\text{Pt}}$ at -0.07 V vs. RHE (**Figure R10d**), respectively. As shown in **Figure R10e**, the Tafel slopes of $\text{Pt}_{21}\text{Ni}_{27}\text{Fe}_{19}\text{Co}_{17}\text{Cu}_{16}/\text{C}$, $\text{Pt}_{18}\text{Ni}_{26}\text{Fe}_{15}\text{Co}_{14}\text{Cu}_{27}/\text{C}$ and $\text{Pt}_{15}\text{Ni}_{24}\text{Fe}_{13}\text{Co}_{13}\text{Cu}_{35}/\text{C}$ are 36 mV dec^{-1} , 30 mV dec^{-1} and 45 mV dec^{-1} , respectively, which demonstrate that the catalyst greatly boosts HER kinetics. The stability of the $\text{Pt}_{21}\text{Ni}_{27}\text{Fe}_{19}\text{Co}_{17}\text{Cu}_{16}/\text{C}$, $\text{Pt}_{18}\text{Ni}_{26}\text{Fe}_{15}\text{Co}_{14}\text{Cu}_{27}/\text{C}$ and $\text{Pt}_{15}\text{Ni}_{24}\text{Fe}_{13}\text{Co}_{13}\text{Cu}_{35}/\text{C}$ catalysts were measured at overpotential of 12 mV, 11 mV and 15 mV (under current density 10 mA cm^{-2} condition, **Figure R10f**), respectively. We can see that the stability of the $\text{Pt}_{18}\text{Ni}_{26}\text{Fe}_{15}\text{Co}_{14}\text{Cu}_{27}/\text{C}$ catalyst is higher than $\text{Pt}_{21}\text{Ni}_{27}\text{Fe}_{19}\text{Co}_{17}\text{Cu}_{16}/\text{C}$ and $\text{Pt}_{15}\text{Ni}_{24}\text{Fe}_{13}\text{Co}_{13}\text{Cu}_{35}/\text{C}$ catalysts.

Figure R11. (a) and (b) CV curves in the double layer region at scan rates of 20, 40, 60, 80 and 100 mV s^{-1} in 1.0 M KOH for $\text{Pt}_{21}\text{Ni}_{27}\text{Fe}_{19}\text{Co}_{17}\text{Cu}_{16}/\text{C}$ and $\text{Pt}_{15}\text{Ni}_{24}\text{Fe}_{13}\text{Co}_{13}\text{Cu}_{35}/\text{C}$, (d) and (e) Current density as a function of scan rate derived from (a) and (b) at 1.0 V (vs. RHE) of $\text{Pt}_{21}\text{Ni}_{27}\text{Fe}_{19}\text{Co}_{17}\text{Cu}_{16}/\text{C}$ and $\text{Pt}_{15}\text{Ni}_{24}\text{Fe}_{13}\text{Co}_{13}\text{Cu}_{35}/\text{C}$. (c) CV curves of in $0.5 \text{ M H}_2\text{SO}_4$ and (f) The potential dependent TOF curves of $\text{Pt}_{21}\text{Ni}_{27}\text{Fe}_{19}\text{Co}_{17}\text{Cu}_{16}/\text{C}$, $\text{Pt}_{18}\text{Ni}_{26}\text{Fe}_{15}\text{Co}_{14}\text{Cu}_{27}/\text{C}$ and $\text{Pt}_{15}\text{Ni}_{24}\text{Fe}_{13}\text{Co}_{13}\text{Cu}_{35}/\text{C}$.

We found that the electrochemical double-layer capacitance of Pt₁₈Ni₂₆Fe₁₅Co₁₄Cu₂₇/C catalyst is higher than Pt₂₁Ni₂₇Fe₁₉Co₁₇Cu₁₆/C and Pt₁₅Ni₂₄Fe₁₃Co₁₃Cu₃₅/C (**Figure R11** and **Supplementary Figure 5**), indicating that the Pt₁₈Ni₂₆Fe₁₅Co₁₄Cu₂₇/C can expose more active sites. The TOF value is used to characterize the activity of each site in the catalyst. It is found that the TOF value of the Pt₁₈Ni₂₆Fe₁₅Co₁₄Cu₂₇/C catalyst is also higher than Pt₂₁Ni₂₇Fe₁₉Co₁₇Cu₁₆/C and Pt₁₅Ni₂₄Fe₁₃Co₁₃Cu₃₅/C catalysts under various potentials (**Figure R11f**), which shows the faster HER kinetics of Pt₁₈Ni₂₆Fe₁₅Co₁₄Cu₂₇/C catalyst. These results further indicate that the Pt₁₈Ni₂₆Fe₁₅Co₁₄Cu₂₇/C can improve the catalytic activity effectively.

Figure R12. (a) and (b) HER polarization curves (geometrical area) for Pt₂₁Ni₂₇Fe₁₉Co₁₇Cu₁₆/C and Pt₁₅Ni₂₄Fe₁₃Co₁₃Cu₃₅/C after 10000 CV cycle. (c) The Nyquist plots at -50 mV vs. RHE of Pt₂₁Ni₂₇Fe₁₉Co₁₇Cu₁₆/C, Pt₁₈Ni₂₆Fe₁₅Co₁₄Cu₂₇/C and Pt₁₅Ni₂₄Fe₁₃Co₁₃Cu₃₅/C.

To further evaluate the stability of Pt₂₁Ni₂₇Fe₁₉Co₁₇Cu₁₆/C, Pt₁₈Ni₂₆Fe₁₅Co₁₄Cu₂₇/C and Pt₁₅Ni₂₄Fe₁₃Co₁₃Cu₃₅/C catalysts, the 10000th CV cycles in 1 M KOH solution were performed. **Figure R12 a** and **b** display the LSV curves of Pt₂₁Ni₂₇Fe₁₉Co₁₇Cu₁₆/C and Pt₁₅Ni₂₄Fe₁₃Co₁₃Cu₃₅/C catalysts before and after 10000th CV cycles. The stability of the Pt₂₁Ni₂₇Fe₁₉Co₁₇Cu₁₆/C and Pt₁₅Ni₂₄Fe₁₃Co₁₃Cu₃₅/C catalysts decrease slightly. Also, electrochemical impedance spectra (**Figure R12c**) exhibits Pt₁₈Ni₂₆Fe₁₅Co₁₄Cu₂₇/C catalyst has a smaller semicircular diameter than Pt₂₁Ni₂₇Fe₁₉Co₁₇Cu₁₆/C and Pt₁₅Ni₂₄Fe₁₃Co₁₃Cu₃₅/C catalyst, the transfer resistance of Pt₁₈Ni₂₆Fe₁₅Co₁₄Cu₂₇/C is much lower than that of Pt₂₁Ni₂₇Fe₁₉Co₁₇Cu₁₆/C and Pt₁₅Ni₂₄Fe₁₃Co₁₃Cu₃₅/C catalysts, indicating higher interfacial charge transfer rate and faster HER kinetics.

Figure R13. Methanol electro-oxidation performance of the $\text{Pt}_{21}\text{Ni}_{27}\text{Fe}_{19}\text{Co}_{17}\text{Cu}_{16}/\text{C}$, $\text{Pt}_{18}\text{Ni}_{26}\text{Fe}_{15}\text{Co}_{14}\text{Cu}_{27}/\text{C}$ and $\text{Pt}_{15}\text{Ni}_{24}\text{Fe}_{13}\text{Co}_{13}\text{Cu}_{35}/\text{C}$ in 1 M KOH + 1 M CH_3OH electrolyte. (a) CV curves (area activity). (b) CV curves (mass activity). (c) Peak values of mass activity and area activity. (d) and (e) CV curves of the $\text{Pt}_{21}\text{Ni}_{27}\text{Fe}_{19}\text{Co}_{17}\text{Cu}_{16}/\text{C}$ and $\text{Pt}_{15}\text{Ni}_{24}\text{Fe}_{13}\text{Co}_{13}\text{Cu}_{35}/\text{C}$ before and after 1000 cycles. (f) Chronoamperometric tests for MOR at 0.65 V vs. RHE.

As shown in **Figure R13a** and **b**, the $\text{Pt}_{18}\text{Ni}_{26}\text{Fe}_{15}\text{Co}_{14}\text{Cu}_{27}/\text{C}$ catalyst shows higher activity compared to $\text{Pt}_{21}\text{Ni}_{27}\text{Fe}_{19}\text{Co}_{17}\text{Cu}_{16}/\text{C}$ and $\text{Pt}_{15}\text{Ni}_{24}\text{Fe}_{13}\text{Co}_{13}\text{Cu}_{35}/\text{C}$ catalysts for MOR in 1 M KOH + 1 M CH_3OH electrolyte at a sweep rate of 20 mV s^{-1} . The $\text{Pt}_{18}\text{Ni}_{26}\text{Fe}_{15}\text{Co}_{14}\text{Cu}_{27}/\text{C}$ catalyst in mass activity and area activity ($15.04 \text{ A mg}^{-1}_{\text{Pt}}$, $114.93 \text{ mA cm}^{-2}$) are higher than that of $\text{Pt}_{21}\text{Ni}_{27}\text{Fe}_{19}\text{Co}_{17}\text{Cu}_{16}/\text{C}$ ($14.16 \text{ A mg}^{-1}_{\text{Pt}}$, $110.25 \text{ mA cm}^{-2}$) and $\text{Pt}_{15}\text{Ni}_{24}\text{Fe}_{13}\text{Co}_{13}\text{Cu}_{35}/\text{C}$ ($13.85 \text{ A mg}^{-1}_{\text{Pt}}$, $105.87 \text{ mA cm}^{-2}$) at peak potential for MOR (**Figure R13c**).

To study the stability of the $\text{Pt}_{18}\text{Ni}_{26}\text{Fe}_{15}\text{Co}_{14}\text{Cu}_{27}/\text{C}$ and Pt/C catalysts in MOR, the chronoamperometry test at 0.65 V vs. RHE and 1000 CV cycles were executed. After 1000 CV cycles, the mass activity of the $\text{Pt}_{21}\text{Ni}_{27}\text{Fe}_{19}\text{Co}_{17}\text{Cu}_{16}/\text{C}$ and $\text{Pt}_{15}\text{Ni}_{24}\text{Fe}_{13}\text{Co}_{13}\text{Cu}_{35}/\text{C}$ catalysts decay by about 6.5% and 12.9% (**Figure 3d** and **e**), further confirmed the excellent stability of $\text{Pt}_{18}\text{Ni}_{26}\text{Fe}_{15}\text{Co}_{14}\text{Cu}_{27}/\text{C}$ catalyst (about 6.4%). After 5000 s chronoamperometry test (**Figure R13f**), the $\text{Pt}_{18}\text{Ni}_{26}\text{Fe}_{15}\text{Co}_{14}\text{Cu}_{27}/\text{C}$ catalyst shows higher stability than that of $\text{Pt}_{21}\text{Ni}_{27}\text{Fe}_{19}\text{Co}_{17}\text{Cu}_{16}/\text{C}$ and $\text{Pt}_{15}\text{Ni}_{24}\text{Fe}_{13}\text{Co}_{13}\text{Cu}_{35}/\text{C}$ catalysts.

Figure R14. (a) and (b) The TEM images of $\text{Pt}_{21}\text{Ni}_{27}\text{Fe}_{19}\text{Co}_{17}\text{Cu}_{16}/\text{C}$ after stability test for HER and MOR. (c) and (d) The TEM images of $\text{Pt}_{15}\text{Ni}_{24}\text{Fe}_{13}\text{Co}_{13}\text{Cu}_{35}/\text{C}$ after stability test for HER and MOR.

The morphologies of $\text{Pt}_{21}\text{Ni}_{27}\text{Fe}_{19}\text{Co}_{17}\text{Cu}_{16}/\text{C}$ and $\text{Pt}_{15}\text{Ni}_{24}\text{Fe}_{13}\text{Co}_{13}\text{Cu}_{35}/\text{C}$ catalyst do not obvious change after HER stability test (**Figure 14a** and **c**). After MOR stability test, from the **Figure 14b** and **d**, we can see that the NPs are slightly aggregated.

Figure R15. The CO stripping curves for (a) $\text{Pt}_{21}\text{Ni}_{27}\text{Fe}_{19}\text{Co}_{17}\text{Cu}_{16}/\text{C}$ and (b) $\text{Pt}_{15}\text{Ni}_{24}\text{Fe}_{13}\text{Co}_{13}\text{Cu}_{35}/\text{C}$ in 1 M KOH electrolyte.

And from CO stripping curves (**Figure R15** and **Supplementary Figure 32**), the onset potentials of the $\text{Pt}_{18}\text{Ni}_{26}\text{Fe}_{15}\text{Co}_{14}\text{Cu}_{27}/\text{C}$ (0.391 V vs. RHE) display 4 mV and 22 mV decrease compared with $\text{Pt}_{21}\text{Ni}_{27}\text{Fe}_{19}\text{Co}_{17}\text{Cu}_{16}/\text{C}$ (0.395 V vs. RHE) and $\text{Pt}_{15}\text{Ni}_{24}\text{Fe}_{13}\text{Co}_{13}\text{Cu}_{35}/\text{C}$ (0.413 V vs. RHE) catalyst. It further shows that the $\text{Pt}_{18}\text{Ni}_{26}\text{Fe}_{15}\text{Co}_{14}\text{Cu}_{27}/\text{C}$ catalyst has better CO anti-poisoning performance.

Manuscript Revision: The related material preparation, characterization and catalytic performance discussion have been added to the revised manuscript. (please see **4-23 lines** (the red-label part) of **Page 8**, **1-2** and **12-15 lines** (the red-label part) of **Page 9**, **1-3** and **15-17 lines** (the red-label part) of **Page 10**, in the revised manuscript. **10-18 lines** (the red-label part) of **Page 2** in the Supporting Information). The **Figure R2-R7** has been given in the Supporting Information (please see **Supplementary Figure 14-24**, **Supplementary Figure 26-28**, **Supplementary Figure 30**, **Supplementary Figure 32**, **Supplementary Table 5** and **Supplementary Table 6**).

Question 4: Does the stoichiometry stay constant under (prolonged) HER operation? Supplementary table points to that, but can the slight variations not be caused by leaching? Please check by preferably ICP-MS of the spent electrolyte.

Author Reply 4: Thank you for reviewer's suggestion. We performed ICP test for the spent electrolyte as suggested by the reviewer. After the stability test, the ICP results showed that the non-noble metal components (Ni Fe Co Cu) decreased, indicating that Ni, Fe, Co, and Cu atoms leached from the catalyst surface. To verify this result, the ICP measurement of the solution after durability testing was also conducted. Before the electrocatalytic test, the metal elements contained in the catalyst were not detected in the electrolyte. After the electrolyte is collected and concentrated, the volume is fixed with 5% nitric acid solution for ICP test. It can be seen from the ICP test results (**Table R1**) that the leaching of non-noble metals (Ni, Fe, Co and Cu) in the catalyst. However, Pt was not detected in the electrolyte, it may be that the leaching amount of Pt may be extremely low, which is lower than the detection limit of the instrument. This also shows the excellent stability of the catalyst. [*Chem. Commun.*, 2019, 55, 14526-14529. *Nano Energy*, 2018, 47, 1–7.]

Table R1. The ICP data (mg L⁻¹) of the electrolyte after electro-catalysis test.

Solution	Pt	Ni	Fe	Co	Cu
After HER	-	0.0043	0.0101	0.0089	0.0052
After MOR	-	0.0067	0.0152	0.0076	0.0086

Manuscript Revision: The corresponding sentence “In addition, the slight variations in the stoichiometry (**Supplementary Table 2**) after the electrocatalytic test was caused by the leaching of non-noble metals (Ni, Fe, Co, and Cu) in the catalyst. [*Chem. Commun.*, 2019, 55, 14526-14529. *Nano Energy*, 2018, 47, 1–7.]” has also been added in the revised manuscript (please see **20-22 lines** (the red-label part) of **Page 7** in the revised manuscript). The **Table R1** has been given in the Supporting Information (please see **Supplementary Table 2**).

Reference:

The references (*Chem. Commun.*, 2019, 55, 14526-14529. *Nano Energy*, 2018, 47, 1–7.) were added in the revised manuscript as **Reference 38** and **39** on **Page 21**.

Question 5: The same sensitivity question I have regarding the electronic structure calculations: Did the authors check different random structures and slightly stoichiometry variations to check sensitivity of the PDOS distribution?

Author Reply 5: Thanks for your kind comments. At the preliminary stage of this model, we have compared several different models with slightly different random structures, in which the present model applied in the manuscript has been the most stable one. Thus, we have decided to apply this model with slightly Ni and Cu rich on the surface as our investigation sample. For the density of states, it is mathematically represented as a distribution by a probability density function, and it is generally an average over space and time domains of the various states occupied by the system. Meanwhile, we have also compared the PDOS of the HEA with slightly different stoichiometry. As shown in **Figure R16**, the comparison models show the highly similar electronic structure with the models applied in our work, in which the peak positions and patterns of d-orbitals in each element display very limited change. Therefore, we think that slight stoichiometry variations of the HEA will not significantly affect the electronic structure of the PDOS results. More importantly, the electronic structure of the HEA is more related to the surrounding elements and the coordination environments. Therefore, in our DFT calculations, we have first determined the most stable structure under the same stoichiometry and then we carefully investigate the overall PDOS and site-to-site PDOS of each metal element in the structure to confirm their role in promoting the MOR and HER performances.

Figure R16. The PDOS comparison of the slight stoichiometry variations in HEA structures. (a) The PDOS of applied HEA model in our work. (b)–(d) The PDOS of the different HEA structures with slightly varied stoichiometry.

Manuscript Revision: The corresponding sentence “We have also compared the PDOS of the HEA with slightly different stoichiometry. As shown in **Supplementary Figure 33**, the comparison models show the highly similar electronic structure with the models applied in the work, in which the peak positions and patterns of d-orbitals in each element display very limited change. Therefore, the slight stoichiometry

variations of the HEA will not significantly affect the electronic structure of the PDOS results.” has also been added in the revised manuscript (please see **1-5 lines** (the red-label part) of **Page 11** in the revised manuscript). The **Figure R16** has been given in the Supporting Information (please see **Supplementary Figure 33**).

Question 6: Although the study presents an interesting case of a new catalyst material, there should be more conceptual rationale presented to guide and motivate the choices w.r.t. elements and composition made in this manuscript. Now it is more like a post-factum explanation of the observations by DFT calculations.

Author Reply 6: Thanks very much for reviewer’s comment and suggestion. In the field of electrocatalysis, the high cost of precious metal catalysts severely limits their application in clean energy technology. Therefore, it is very important to develop low-cost, high-durability, and high-activity replacement electrocatalysts to promote the realization of clean energy equipment. Among them, the number of components of the high-entropy alloy can reach five or more, and the number of element combinations is numerous, which can provide many possibilities for adjusting the electronic properties and catalytic activity of the alloy surface.

However, the design space of Pt-based HEA is very large and cannot be fully developed. Therefore, it is necessary to pre-screen the elements alloyed with Pt. In this paper:

(1) Taking into account the high abundance of Ni, Fe, Co, and Cu and they can easily form solid solutions with Pt. These transition metal elements were selected because their similar atomic radius and lower heat of formation make them likely to form stable HEAs. [*Small*, 2019, 1904180. *Sci. China Mater.* 2018, 61, 2–22]

(2) The calculation method provides a means to better understand the catalytic process and direct catalytic design. By calculating the adsorption of molecules and intermediates on the catalyst surface, the DFT can predict the catalytic activity and optimize the composition of HEA, thus becoming a design platform for the discovery of new alloys. At the preliminary stage of this model, we have compared several different models with slightly different random structures, in which the present model applied in the manuscript has been the most stable one. As described in the DFT calculation section of the revised manuscript, the role of each element in the reaction is revealed. The origin of remarkable electroactivity and durability of HEA in HER and MOR is attributed to the synergistic effect of each element for efficient electron transfer. The optimized electronic environment constructs the strategy of the multi-active site for adsorption and transformation of key intermediates, which maximizes the utilization of surface electroactivity.

Based on this, we design and synthesize materials and conduct electrocatalytic testing. Here, Pt and four transition metal elements are controllably incorporated into a single nanostructure phase. In order to uniformly mix these elements in a single-phase nanostructure, the Pt, Ni, Fe, Co, and Cu precursors with equal atomic weight were mixed together, but finally alloy nanoparticles with a molar ratio of $\text{Pt}_{18}\text{Ni}_{26}\text{Fe}_{15}\text{Co}_{14}\text{Cu}_{27}$ were obtained. By changing the content of Cu, alloys with different stoichiometric ratios are obtained, which have high activity to both MOR and HER in alkaline electrolyte.

Through the interaction of different neighboring elements, new active centers are formed, and these new active centers exhibit brand-new properties. The ultimate goal is to adapt these characteristics to any desired response by using almost infinite possible combinations of elements and modifying their composition. Therefore, applications are not necessarily limited to HER and MOR.

Manuscript Revision: The corresponding sentence “The ultimate goal of HEA is to adapt these characteristics to any desired response by using almost infinite possible combinations of elements and modifying their composition.” And “taking into account the high abundance of Ni, Fe, Co, and Cu and they can easily form solid solutions with Pt. These transition metal elements were selected because their similar atomic radius and lower heat of formation make them likely to form stable HEAs. The DFT calculation provides a means to better understand the catalytic process and direct catalytic design.” has also been added in the revised manuscript (please see **10-12 lines, 22-23 lines** (the red-label part) of **Page 3** and **1-2 lines** (the red-label part) of **Page 4** in the revised manuscript).

Question 7: The authors conclude with: “The optimized electronic environment constructs the strategy of the multi-active site for adsorption and transformation of key intermediates, which maximizes the utilization of surface electroactivity.” What does “optimized” mean here? There has been neither design nor optimization involved in the present study - rather an arbitrary composition alloy has been presented here with remarkable activities.

Author Reply 7: Thanks for your comments. We apologize for the confusion caused by our expressions in the manuscript. The “optimized” in this work represent both the structural and electronic optimizations of the HEA nanoparticle synthesized in the experiments. Based on your comments, we have revised the expressions of our work. We aim to explain the origin of the remarkable activities of the synthesized HEA, in which both electronic and structural relaxation have been considered in the DFT calculations. For the structural optimization, we have compared the different random atomic arrangements under the same stoichiometry, in which we notice that the Ni and Cu slight-rich surface show the lowest energies as the optimized structure for the synthesized HEA nanoparticle. This has also been explained in the Methods section. Meanwhile, the electronic structure reveals the synergistic effect of the different metal sites, which guarantees the appropriate binding strength of intermediates for both HER and MOR. Therefore,

the improvements of both lattice stability and electronic structure lead to the optimized HER and MOR performances of the synthesized HEA nanoparticle.

Manuscript Revision: We changed the original sentence to “after a detailed comparison of different random atomic arrangements, the HEA structure with slightly Ni and Cu enriched surface has been applied as the lattice model due to the highest stability.” and “The suitable electronic environment of the HEA realizes the multi-active sites for appropriate adsorption of key intermediates and efficient electron transfer during the electrocatalysis” in the revised manuscript (please see **5-7 lines** (the red-label part) of **Page 11** and **5-6 lines** (the red-label part) of **Page 14** in the revised manuscript).

Question 8: technical details: XPS part: There are oxidized species reported. From the (incomplete) experimental details, it is unclear whether these oxidized metal species (Pt^{2+} , Ni^{2+} , Cu^{2+} ...) arise from the synthesis or from sample handling in air. Please elaborate. XPS under inert atmosphere transfer should be presented. Also, XPS post catalysis should be presented to check for electronic structure changes/oxidation of the materials. Furthermore, XPS experimental details should be updated as suggested in e.g. <http://www.xpsfitting.com/search/label/Publication%20Methods>.

Author Reply 8: Thanks for reviewer’s comment. By consulting the literature, most of the alloys synthesized in the oil phase have some metal oxide species in XPS. The metallic oxide peak might be generated from superficial oxide species. Then, we prepared the material in inert atmosphere (the experiment process is operated under Ar atmosphere) and carried out the XPS test. The XPS analyses were carried out with Axis Supra spectrometer using a monochromatic Al K α source (15mA, 14kV). Survey scan analyses were carried out with an analysis area of 300 x 700 microns and a pass energy of 100 eV. High resolution analyses were carried out with an analysis area of 300 x 700 microns and a pass energy of 30 eV. Spectra have been corrected to the main line of the carbon 1s spectrum (adventitious carbon) set to 284.8 eV. Spectra were analysed using CasaXPS software (version 2.3.14).

From **Figure R17** and **Supplementary Figure 2** in the supporting information, we can see that in an inert atmosphere, there are still oxide species. Compared with the material under normal conditions, the oxide species are reduced, especially the Ni oxide species is greatly reduced.

Figure R17. (a) Pt 4f, (b) Ni 2p, (c) Fe 2p, (d) Co 2p, and (e) Cu 2p XPS spectrum of Pt₁₈Ni₂₆Fe₁₅Co₁₄Cu₂₇/C catalyst under inert atmosphere. (f) The valence state ratios of Pt⁰/Pt²⁺, Ni⁰/Ni²⁺, Fe⁰/Fe²⁺, Co⁰/Co²⁺, and Cu⁰/Cu²⁺ obtained from XPS spectra of Pt₁₈Ni₂₆Fe₁₅Co₁₄Cu₂₇/C catalyst.

Figure R18. (a) Pt 4f, (b) Ni 2p, (c) Fe 2p, (d) Co 2p, and (e) Cu 2p XPS spectrum of Pt₁₈Ni₂₆Fe₁₅Co₁₄Cu₂₇/C catalyst after HER.

In addition, we added the XPS spectrum after the electrocatalytic test. XPS analysis (**Figure R18** and **R19**) shows that the Pt 4f, Ni 2p, Fe 2p, Co 2p and Cu 2p spectra of Pt₁₈Ni₂₆Fe₁₅Co₁₄Cu₂₇/C were very similar before and after electrocatalysis, while the oxide species increased slightly (**Table R2**).

Figure R19. (a) Pt 4f, (b) Ni 2p, (c) Fe 2p, (d) Co 2p, and (e) Cu 2p XPS spectrum of Pt₁₈Ni₂₆Fe₁₅Co₁₄Cu₂₇/C catalyst after MOR.

Table R2. The valence state ratios of Pt⁰/Pt²⁺, Ni⁰/Ni²⁺, Fe⁰/Fe²⁺, Co⁰/Co²⁺, and Cu⁰/Cu²⁺ obtained from XPS spectra of Pt₁₈Ni₂₆Fe₁₅Co₁₄Cu₂₇/C catalyst.

Catalysts	Pt ⁰ /Pt ²⁺	Ni ⁰ /Ni ²⁺	Fe ⁰ /Fe ²⁺	Co ⁰ /Co ²⁺	Cu ⁰ /Cu ²⁺
Initial	1.54	0.19	5.11	1.15	4.45
After HER	1.30	0.15	2.86	1.14	4.14
After MOR	1.52	0.16	3.01	1.11	4.20

Manuscript Revision: In **Characterization** section, we have added a brief description about the XPS experimental details, such as “The XPS analyses were carried out with Axis Supra spectrometer using a monochromatic Al K α source (15mA, 14kV). Survey scan analyses were carried out with an analysis area of 300 x 700 microns and a pass energy of 100 eV. High resolution analyses were carried out with an analysis area of 300 x 700 microns and a pass energy of 30 eV. Spectra have been corrected to the main line of the carbon 1s spectrum (adventitious carbon) set to 284.8 eV. Spectra were analysed using

CasaXPS software (version 2.3.14).” (Please see **7-13 lines** (the red-label part) of **Page 15** in the revised manuscript).

And “XPS analysis (**Supplementary Figure 12**) shows that the Pt 4f, Ni 2p, Fe 2p, Co 2p and Cu 2p spectra of Pt₁₈Ni₂₆Fe₁₅Co₁₄Cu₂₇/C were very similar before and after electrocatalysis, while the oxide species increased slightly (**Supplementary Table 3**).” “XPS spectrum (**Supplementary Figure 31**) shows that the Pt 4f, Ni 2p, Fe 2p, Co 2p and Cu 2p spectra of Pt₁₈Ni₂₆Fe₁₅Co₁₄Cu₂₇/C were very similar before and after MOR electrocatalysis.” has also been added in the revised manuscript (please see **22-23 lines** (the red-label part) of **Page 7** in the revised manuscript and **6-8 lines** (the red-label part) of **Page 10** in the revised manuscript).

The **Figure R18**, **Figure R19** and **Table R2** has been given in the Supporting Information (please see **Supplementary Figure 12**, **Supplementary Figure 31** and **Supplementary Table 3**).

To Reviewer 2:

General Comment: This work reports the synthesis, characterization and reactivity towards electrocatalytic performance for hydrogen evolution reaction (HER) and methanol oxidation reaction (MOR) of Pt₁₈Ni₂₆Fe₁₅Co₁₄Cu₂₇/C NPs. Different techniques are used, experimental and theoretical. The methods used are state-of-the-art and well chosen. The paper claims to discovery of a particular efficient en selective electro-catalyst, which is to my option justified. Moreover, the excellent work is supported by pertinent results and will be of no doubt of great interest to others in the community. The research is in line with the investigation strategy based on multi-metal alloys (more than 3 metals).

I have only one comment that the authors might consider in their future works: This very complex question is particularly well suited to be studied with the statistical machine learning simulation methods, and would be a perfect example to be tested.

The chemical space (5 metals) is particularly challenging.

As far as the DFT calculations are concerned I want to say that the analysis of the electronic properties (PDOS, band gaps and orbital overlaps, ...) are very pedagogically presented and well interpreted, which is important for a the readability of a wide public.

The MS is well written and the graphs and figures are clear.

I would like to congratulate the authors for their nice work.

The work can be published after introducing some minor changes (see above and below).

Author Reply: Thank you for your precious support and appreciation of our work. Your kind comments are very supportive to our present work and our future works in this field. We sincerely hope that our work will deliver a novel research to the electrocatalyst design in the future.

Small comments:

I suggest the authors to precise the term "DFT" by completing it to "Periodic DFT".

Author Reply: Thanks for your comments. We have revised the DFT in our calculation to "Periodic DFT" as suggested by the Reviewer.

To Reviewer 3

General Comment: This work reports a new synthesis route for Pt-containing HEA nanoparticles which showed excellent HER and MOR activities and stability. The performance was explained with the help of DFT calculations which suggest synergistic effect from multiple surface sites for the two reactions. The work has potential to interest broad field of readers. But a few issues need be addressed before its acceptance for publication in *Nat. Commun.*

Author Reply: Thank you for the valuable confirmation of our work. We highly appreciate your efforts in reviewing our work and giving valuable comments. Based on your valuable comments, we have supplied the detailed.

Question 1: In the DFT studies, the authors need to explain why Ni and Cu are slightly enriched on the surface. According to the XPS result of the fresh sample, it appears Pt should be enriched on the surface. Actually XPS spectra of the used samples after HER and MOR should be collected to analyze the surface composition change. It appears from the ICP analysis (Table S1) that Pt relative ratio increases after the two reactions. For more accurate analysis of the surface composition, XPS is not recommended unless energy-scanned XPS at a synchrotron source is conducted to get more surface sensitive information. Low energy ion scattering is more suited for top surface composition analysis.

Author Reply 1: Thanks for your kind comment. In our preliminary calculations, we have compared the stability of different lattice structures of HEA nanoparticle with slightly different atomic arrangements under the same stoichiometry $\text{Pt}_{18}\text{Ni}_{26}\text{Fe}_{15}\text{Co}_{14}\text{Cu}_{27}$. After carefully comparing the total energy of the lattice structure, we notice that the slightly Cu and Ni enriched surface shows the lowest energies, which has been determined as the structure model in the calculation of this work. We have also explained this in the Methods section. Meanwhile, considering the Pt ratio and XPS results, Pt is also slightly more

enriched near the surface rather than the bulk. We think the slight increases of the Pt ratio originates from the interlayer relaxation after the HER and MOR, leading to a more evident change of Pt. However, a highly precise characterization of each element is still challenging, especially for the HEA structure. Our previous work has systematically investigated the lattice strain in the transition metal electrocatalyst and confirmed the potential of HEA for electrocatalysis [Small, 2020, DOI: 10.1002/sml.202002434].

In this work, we would like to highlight the synthesis of HEA nanoparticle and their superior performances in HER and MOR. Instead, the slight ratio changes of the element are not the key focus of this work. We also agree with the Reviewer that low energy ion scattering is more appropriate is more suited for top surface composition analysis while XPS results supply the valence state information of the surface element. However, the low-energy ion scattering test cannot be performed due to limited conditions. We also know that the surface composition of the catalyst has a great influence on the catalytic performance. We further use Ar⁺ sputtering to study the change in surface element content of particles. [*J. Am. Chem. Soc.* 2017, 139, 12283–12290] After Ar⁺ sputtering for 10 s, the relative content of the elements has been reduced, but the reduced contents of Ni, Cu and Fe were slightly higher than that of Pt and Co (**Table R1**).

Table R1. The atomic ratio of Pt, Ni, Fe, Co, and Cu obtained from XPS after Ar⁺ sputtering for 10 s.

Catalysts	Pt	Ni	Fe	Co	Cu
Initial	21.2	27.4	15.2	13.7	22.5
After Ar ⁺ sputtering	21.4	27.1	15.3	13.9	22.3

In addition, we performed the XPS spectrums of the used samples after HER and MOR as suggested by the reviewer. XPS analysis (**Figure R1** and **R2**) shows that the Pt 4f, Ni 2p, Fe 2p, Co 2p and Cu 2p spectra of Pt₁₈Ni₂₆Fe₁₅Co₁₄Cu₂₇/C were very similar before and after electrocatalysis, while the oxide species increased slightly (**Table R2**).

Table R2. The valence state ratios of Pt⁰/Pt²⁺, Ni⁰/Ni²⁺, Fe⁰/Fe²⁺, Co⁰/Co²⁺, and Cu⁰/Cu²⁺ obtained from XPS spectra of Pt₁₈Ni₂₆Fe₁₅Co₁₄Cu₂₇/C catalyst.

Catalysts	Pt ⁰ /Pt ²⁺	Ni ⁰ /Ni ²⁺	Fe ⁰ /Fe ²⁺	Co ⁰ /Co ²⁺	Cu ⁰ /Cu ²⁺
Initial	1.54	0.19	5.11	1.15	4.45
After HER	1.30	0.15	2.86	1.14	4.14
After MOR	1.52	0.16	3.01	1.11	4.20

Figure R1. (a) Pt 4f, (b) Ni 2p, (c) Fe 2p, (d) Co 2p, and (e) Cu 2p XPS spectrum of $\text{Pt}_{18}\text{Ni}_{26}\text{Fe}_{15}\text{Co}_{14}\text{Cu}_{27}/\text{C}$ catalyst after HER.

Figure R2. (a) Pt 4f, (b) Ni 2p, (c) Fe 2p, (d) Co 2p, and (e) Cu 2p XPS spectrum of $\text{Pt}_{18}\text{Ni}_{26}\text{Fe}_{15}\text{Co}_{14}\text{Cu}_{27}/\text{C}$ catalyst after MOR.

Manuscript Revision: The corresponding sentence “It is well known that the surface composition of the catalyst affects the catalytic performance. We use Ar⁺ sputtering to study the change in surface element content of the PtNiFeCoCu NPs. [*J. Am. Chem. Soc.* 2017, 139, 12283–12290] After Ar⁺ sputtering for 10 s, the relative content of the elements has been reduced, but the reduced contents of Ni, Cu and Fe were slightly higher than that of Pt and Co (**Supplementary Table 4**).” has also been added in the revised manuscript (please see **20-23 lines** (the red-label part) of **Page 10** in the revised manuscript). And “XPS analysis (**Supplementary Figure 12**) shows that the Pt 4f, Ni 2p, Fe 2p, Co 2p and Cu 2p spectra of Pt₁₈Ni₂₆Fe₁₅Co₁₄Cu₂₇/C were very similar before and after electrocatalysis, while the oxide species increased slightly (**Supplementary Table 3**).” “XPS spectrum (**Supplementary Figure 31**) shows that the Pt 4f, Ni 2p, Fe 2p, Co 2p and Cu 2p spectra of Pt₁₈Ni₂₆Fe₁₅Co₁₄Cu₂₇/C were very similar before and after MOR electrocatalysis.” has also been added in the revised manuscript (please see **22-23 lines** (the red-label part) of **Page 7** in the revised manuscript and **6-8 lines** (the red-label part) of **Page 10** in the revised manuscript).

The **Figure R1, Figure R2, Table R1 and Table R2** has been given in the Supporting Information (please see **Supplementary Figure 12, Supplementary Figure 31, Supplementary Table 4 and Supplementary Table 3**).

Reference:

The reference (*J. Am. Chem. Soc.* 2017, 139, 12283–12290) was added in the revised manuscript as **Reference 43** on **Page 22**.

Question 2: More experimental details should be provided for the synthesis of the HEAs and HEAs/C samples. is there any surfactant left on the surface even after the activation before the electrocatalysis tests?

Author Reply 2: Thanks for reviewer’s valuable comment and suggestion. We have added the experimental details for the synthesis of HEAs and HEAs/C samples. As follows:

“Preparation of high-entropy alloy Pt₁₈Ni₂₆Fe₁₅Co₁₃Cu₂₇ Nanoparticles (NPs). CTAC (50 mg) was added into oleylamine (5 mL) in a 15 mL vial. After sonication for about 15 min, Pt(acac)₂ (10 mg), Ni(acac)₂ (6.4 mg), Fe(acac)₃ (8.8 mg), Co(acac)₃ (8.9 mg), Cu(acac)₂ (6.5 mg), glucose (60 mg) and Mo(CO)₆ (33 mg) were added into the vial. In order to obtain a homogeneous solution, the mixture was sonicated for 1 h. The vial was heated to 220 °C at 5 °C min⁻¹ and then kept 2 h under magnetic stirring at 400 rpm. The black colloidal products were collected by centrifugation and washed two times with an ethanol/cyclohexane mixture. Finally, the black colloidal products were kept in cyclohexane for further use. (**Please see the Methods section in the revised manuscript**)

Preparation of PtNiFeCoCu/C. The obtained 1 mg HEA NPs dispersed in 10 mL cyclohexane was mixed with 4 mg of carbon (Ketjen Black-300) in 10 mL ethanol under sonication for 1 h, and then the product was collected via centrifugation with ethanol. The PtNiFeCoCu/C catalysts were further cleaned

(remove organic species) with 0.5 M acetic acid (ethanol solution) under N₂ atmosphere. After being sonicating for 2 h, the products were collected by centrifugation and washed with ethanol for three times. **(Please see the Materials and Methods section in the Supporting Information)**

To investigate the catalytic performance of the HEAs/C for HER and MOR. Before electrochemical tests, we removed oleylamine and CTAC from the HEAs by washing them with an ethanol/cyclohexane mixture many times and treated with acetic acid to further remove the residual surfactant. To prove this result, we characterized the sample by FTIR spectroscopy (**Figure R3**). [*Nano Lett.* 2020, 20, 1967–1973] After washing them with an ethanol/cyclohexane mixture, it can be seen that some weak peaks, appeared in the FT-IR spectra of the PtNiFeCoCu/C (Magenta line in the **Figure R3**), suggesting the small amount of residual adsorption of oleylamine and CTAC on the surface of PtNiFeCoCu/C.

After acetic acid treatment (the purified PtNiFeCoCu/C, olive line in the **Figure R3**), no obvious characteristic peaks corresponding to oleylamine and CTAC were observed in the FTIR spectrum of HEAs. After further activation (Navy line in the **Figure R3**), the characteristic peaks of CTAC and oleylamine were not observed in the FTIR spectrum.

Figure R3. FTIR spectra of the pure KBr, pure oleylamine, pure CTAC, PtNiFeCoCu/C, the purified PtNiFeCoCu/C and after the activation before the electrocatalysis tests.

Manuscript Revision: In the method section, we changed the original sentence to “**Preparation of high-entropy alloy Pt₁₈Ni₂₆Fe₁₅Co₁₃Cu₂₇ Nanoparticles (NPs).** CTAC (50 mg) was added into oleylamine (5 mL) in a 15 mL vial. After sonication for about 15 min, Pt(acac)₂ (10 mg), Ni(acac)₂ (6.4 mg), Fe(acac)₃ (8.8 mg), Co(acac)₃ (8.9 mg), Cu(acac)₂ (6.5 mg), glucose (60 mg) and Mo(CO)₆ (33 mg)

were added into the vial. In order to obtain a homogeneous solution, the mixture was sonicated for 1 h. The vial was heated to 220 °C at 5 °C min⁻¹ and then kept 2 h under magnetic stirring at 400 rpm. The black colloidal products were collected by centrifugation and washed two times with an ethanol/cyclohexane mixture. Finally, the black colloidal products were kept in cyclohexane for further use.” (Please see the **Methods section in the revised manuscript**) And “**Preparation of PtNiFeCoCu/C**. The obtained 1 mg HEA NPs dispersed in 10 mL cyclohexane was mixed with 4 mg of carbon (Ketjen Black-300) in 10 mL ethanol under sonication for 1 h, and then the product was collected via centrifugation with ethanol. The PtNiFeCoCu/C catalysts were further cleaned (remove organic species) with 0.5 M acetic acid (ethanol solution) under N₂ atmosphere. After being sonicating for 2 h, the products were collected by centrifugation and washed with ethanol for three times.” (Please see the **Materials and Methods section in the Supporting Information**)

The corresponding sentence “In order to study the electrocatalytic performance of PtNiFeCoCu/C, we further treated with acetic acid to remove residual surfactants. From the FTIR spectrum (**Supplementary Figure 3**), the acetic acid-treated sample (the purified PtNiFeCoCu/C), no obvious characteristic peaks corresponding to oleylamine and CTAC were observed (*Nano Lett.* 2020, 20, 1967–1973).” has also been added in the revised manuscript (please see **21-23 lines** (the red-label part) of **Page 5** and **1-2 lines** (the red-label part) of **Page 6** in the revised manuscript), and the **Figure R3** has been given in the Supporting Information (please see **Supplementary Figure 3**).

Reference:

The reference (*Nano Lett.* 2020, 20, 1967–1973) was added in the revised manuscript as **Reference 36** on **Page 21**.

Question 3: Did the authors measure the Pt size on the Pt/C sample? I could not seem to find the information in the paper. The Pt size should be at least similar to that of the HEA particles so the electrocatalytic activity can be reasonably compared based on surface Pt sites.

Author Reply 3: Thanks for reviewer’s valuable comment and suggestion. We have measured the Pt size on the Pt/C sample (about 3.0 nm, **Figure R4**), it is similar to the size of HEA particles (about 3.4 nm, **Figure 1a** in the revised manuscript).

Figure R4. (a) The TEM image and (b) histogram of the diameter of Pt/C catalyst.

Manuscript Revision: The corresponding sentence “The Pt size (about 3.0 nm, **Supplementary Figure 4**) on the Pt/C catalyst is similar to the size of HEA particles, so the electrocatalytic activity can be reasonably compared based on surface Pt sites.” has also been added in the revised manuscript (please see **3-5 lines** (the red-label part) of **Page 6** in the revised manuscript), and the **Figure R4** has been given in the Supporting Information (please see **Supplementary Figure 4**).

Question 4: A minor note for table S1, the number of digit after the decimal should be consistent for all measurements: the fresh sample results have no decimals while the used sample results show 1 digit after the decimal. The same problem is present for tables S2 and S3.

Author Reply 4: Thanks for your comments. We have revised the number of digit in **Supplementary Table 1 (Table S1)**, **Supplementary Table 7 (Table S2)**, and **Supplementary Table 8 (Table S3)** to be the same as suggested by the Reviewer.

In the end, we would like to express our thanks to the precious time of the reviewers and the editor. We sincerely wish that our point-to-point response and the revised manuscript can address your concerns and satisfy your requirements for publications. We would be grateful if we have the chance to share our work with readers of *Nature Communications*.

Sincerely yours,

Lei Wang

REVIEWERS' COMMENTS

Reviewer #1 (Remarks to the Author):

The authors have appropriately addressed the requests and concerns of the three reviewers in a very detailed and thorough manner.

Publication of the manuscript can be recommended in the present form.

Reviewer #3 (Remarks to the Author):

The authors have addressed my previous comments. I recommend its acceptance of current version.

Reply to Reviewers' Comments

Dear Reviewers,

Thank you for your precious time to constructive comments on our manuscript titled “**Fast Site-to-site Electron Transfer of High-entropy Alloy Nanocatalyst Driving Redox Electrocatalysis**” (Manuscript ID: NCOMMS-20-29577A) for *Nature Communications*. We sincerely appreciate your opinions and confirmation of our work.

To Reviewer 1:

General Comment: The authors have appropriately addressed the requests and concerns of the three reviewers in a very detailed and thorough manner.

Publication of the manuscript can be recommended in the present form.

Author Reply: Thanks for the comment. We are delight to know that you were satisfied with the revisions.

To Reviewer 3:

General Comment: The authors have addressed my previous comments. I recommend its acceptance of current version.

Author Reply: Many thanks for the kind comment.

We would like to thank all referees who are satisfied with this revision and recommend that the paper be accepted for publication in *Nature Communications*.

Sincerely yours,

Lei Wang